



# Inferring the anthropogenic NOₓ emission trend over the United States during 2003 - 2017 from satellite observations: Was there a flattening of the emission tend after the Great Recession?

Jianfeng Li[1], Yuhang Wang[1*]

[1] School of Earth and Atmospheric Sciences, Georgia Institute of Technology, Atlanta, Georgia, USA

[*] *Correspondence to* Yuhang Wang (yuhang.wang@eas.gatech.edu)





## 11 **Abstract**

We illustrate the nonlinear relationships among anthropogenic $NO_x$ emissions, $NO_2$
tropospheric vertical column densities (TVCDs), and $NO_2$ surface concentrations using model
simulations for July 2011 over the contiguous United States (CONUS). The variations of $NO_2$
surface concentrations and TVCDs are generally consistent and reflect well anthropogenic $NO_x$
emission variations for high-anthropogenic-$NO_x$ emission regions. For low-anthropogenic-$NO_x$
emission regions, however, nonlinearity in the emission-TVCD relationship makes it difficult to
use satellite observations to infer anthropogenic $NO_x$ emission changes. The analysis is extended
to 2003 – 2017. Similar variations of $NO_2$ surface measurements and coincident satellite $NO_2$
TVCDs over urban regions are in sharp contrast to the large variation differences between surface
and satellite observations over rural regions. We find a continuous decrease of anthropogenic
$NO_x$ emissions after 2011 by examining surface and satellite measurements in CONUS urban
regions, but the decreasing rate is lower by 9% - 46% than the pre-2011 period.



# 1. Introduction

Anthropogenic emissions of nitrogen oxides ($NO_x = NO_2 + NO$) adversely affect the environment, not only because of their direct detrimental impacts on human health (Greenberg et al., 2016; Greenberg et al., 2017; Heinrich et al., 2013; Weinmayr et al., 2009), but also their fundamental roles in the formation of ozone, acid rain, and fine particles which are unfavorable to human health, ecosystem stabilities, and climate change (Crouse et al., 2015; Kampa and Castanas, 2008; Myhre et al., 2013; Pandey et al., 2005; Singh and Agrawal, 2007). About 48.8 Tg N $yr^{-1}$ of $NO_x$ are emitted globally from both anthropogenic (77%) and natural (23%) sources, such as fossil fuel combustion, biomass and biofuel burning, soil bacteria, and lightning (Seinfeld and Pandis, 2016). 3.85 Tg N and 0.24 Tg N of anthropogenic and natural $NO_x$, respectively, were emitted from the U.S. in 2014 on the basis of the 2014 National Emission Inventory (NEI2014); vehicle sources and fuel combustions accounted for 93% of the total anthropogenic $NO_x$ emissions (EPA, 2017).

The U.S. anthropogenic $NO_x$ emissions during the 2010s declined dramatically compared to the mid-2000s (EPA, 2018; Xing et al., 2013) due to stricter air quality regulations and emission control technology improvements, such as the phase-in of Tier II vehicles during 2004 – 2009 and the switch of power plants from coal to natural gas (De Gouw et al., 2014; McDonald et al., 2018). The overall reduction (about 30% - 50%) of anthropogenic $NO_x$ emissions from the mid-2000s to the 2010s was corroborated by observed decreasing of vehicle $NO_x$ emission factors, $NO_2$ surface concentrations, nitrate wet deposition flux, and $NO_2$ tropospheric vertical column densities (TVCDs) (Bishop and Stedman, 2015; Li et al., 2018; McDonald et al., 2018; Miyazaki et al., 2017; Russell et al., 2012; Tong et al., 2015). However, the detailed $NO_x$ emission changes after the Great Recession (from December 2007 to June 2009) are highly uncertain. On the one hand, the U.S. Environmental Protection Agency (EPA) estimated that the Great Recession had a





slight impact on the anthropogenic $NO_x$ emission trend, and the anthropogenic $NO_x$ emissions
decreased steadily from 2002 to 2017 (Figure S1), although the emission decrease rate slowed
down by about 20% after 2010 (-5.8% $yr^{-1}$ for 2002 – 2010, and -4.7% $yr^{-1}$ for 2010 – 2017,
Table 1) (EPA, 2018). Fuel-based emission estimates in Los Angeles also showed a steady
decrease of anthropogenic $NO_x$ emissions after 2000 and a small impact of the Great Recession
on anthropogenic $NO_x$ emission decrease trend (Hassler et al., 2016). The continuous decrease of
anthropogenic $NO_x$ emissions was consistent with the ongoing reduction of vehicle emissions
(McDonald et al., 2018). On the other hand, Miyazaki et al. (2017) and Jiang et al. (2018) found
that the U.S. $NO_x$ emissions derived from satellite $NO_2$ TVCDs, including OMI (the Ozone
Monitoring Instrument), SCIAMACHY (SCanning Imaging Absorption SpectroMeter for
Atmospheric CHartography), and GOME-2A (Global Ozone Monitoring Experiment – 2 onboard
METOP-A), were almost flat from 2010 - 2015 and suggested that the decrease of $NO_x$ emissions
was only significant before 2010, which was completely different from the bottom-up and fuel-
based emission estimates.
A complicating factor in inferring anthropogenic $NO_x$ emission trends from the observations
of $NO_2$ surface concentrations and satellite $NO_2$ TVCDs is the nonlinearity in $NO_x$ chemistry (Gu
et al., 2013; Gu et al., 2016). Although the decrease rates of both $NO_2$ surface concentrations and
coincident OMI $NO_2$ TVCDs slowed down after the Great Recession over the United States,
Tong et al. (2015), Lamsal et al. (2015) and Jiang et al. (2018) found that the slowdown of the
decrease rates derived from $NO_2$ surface concentrations is 12% - 79% less than those of $NO_2$
TVCDs (Table 1). Secondly, the slowdown of the decrease rates of $NO_2$ surface concentrations
and OMI TVCDs over cities and power plants (Russell et al., 2012; Tong et al., 2015) is
significantly less than those over the whole contiguous United States (CONUS) (Jiang et al.,
2018; Lamsal et al., 2015). Moreover, Zhang et al. (2018) found that filtering out lightning-



affected measurements could significantly improve the comparison of $NO_2$ surface concentration
and OMI $NO_2$ TVCD trends over the CONUS.

In this study, we carefully investigate the relationships among anthropogenic $NO_x$ emissions,

$NO_2$ surface concentrations, and $NO_2$ TVCDs over the CONUS and evaluate the impact of the
relationships on inferring anthropogenic $NO_x$ emission changes and trends from surface and
satellite observations. Section 2 describes the model and datasets used in this study, including the
Regional chEmistry and trAnsport Model (REAM), the EPA Air Quality System (AQS) $NO_2$
surface observations, and $NO_2$ TVCD products from OMI, GOME-2A, GOME-2B (GOME2
onboard METOP-B), and SCIAMACHY. In Section 3, we examine the nonlinear relationships
among anthropogenic $NO_x$ emissions, $NO_2$ surface concentrations, and $NO_2$ TVCDs using model
simulations. Accounting for the effects of chemical nonlinearity, we then investigate the
anthropogenic $NO_x$ emission trends and changes from 2003 – 2017 over the CONUS. Finally,
section 4 gives a summary of the study.
## 2. Model and Data Description
### 2.1 REAM

The REAM model has been applied and evaluated in many research applications including

ozone simulation and forecast, emission inversion and evaluations, and mechanical studies of
chemical and physical processes (Alkuwari et al., 2013; Cheng et al., 2017; Cheng et al., 2018;
Choi et al., 2008a; Choi et al., 2008b; Gu et al., 2013; Gu et al., 2014; Koo et al., 2012; Liu et al.,
2012; Liu et al., 2014; Wang et al., 2007; Yang et al., 2011; Zhang et al., 2017; Zhang et al.,
2018; Zhang and Wang, 2016; Zhao and Wang, 2009; Zhao et al., 2009a; Zhao et al., 2010).
REAM used in this work has 30 vertical layers in the troposphere, and the horizontal resolution is
$36 \times 36$ km$^2$. The model is driven by meteorology fields from a Weather and Research





96 Forecasting (WRF, version 3.6) model simulation initialized and constrained by the NCEP

97 coupled forecast system model version 2 (CFSv2) products (Saha et al., 2011). The chemistry

98 mechanism is based on GEOS-Chem v11.01 with updated reaction rates and aerosol uptake of

99 isoprene nitrates (Fisher et al., 2016). Chemistry boundary conditions and initializations are from

100 a GEOS-Chem ($2° \times 2.5°$) simulation. Hourly anthropogenic emissions on weekdays are based on

101 the 2011 National Emission Inventory (NEI2011), while weekend anthropogenic emissions are

102 set to be two-thirds of the weekday emissions (Beirle et al., 2003; Choi et al., 2012). Biogenic

103 VOC emissions are estimated using the Model of Emissions of Gases and Aerosols from Nature

104 (MEGAN) v2.10 (Guenther et al., 2012). $NO_x$ emissions from soils are based on the Yienger and

105 Levy (YL) scheme (Li et al., 2019; Yienger and Levy, 1995).

106 **2.2 Satellite $NO_2$ TVCDs**

107 In this study, we use $NO_2$ TVCD products from four satellite measurements in the past

108 decade, including SCIAMACHY, GOME-2A, GOME-2B, and OMI, the spectrometers onboard

109 sun-synchronous satellites to monitor atmospheric trace gases. The SCIAMACHY onboard the

110 Environmental Satellite (ENVISAT) has an equator overpass time of 10:00 Local time (LT) and a

111 nadir pixel resolution of $60 \times 30$ km$^2$. The GOME-2 instruments on Metop-A (named as GOME-

112 2A) and Metop-B (GOME-2B) satellites cross the equator at 9:30 LT and have a nadir resolution

113 of $80 \times 40$ km$^2$. After July 15, 2013, the nadir resolution of GOME-2A became $40 \times 40$ km$^2$ with

114 a smaller scanning swath. The OMI onboard the EOS-Aura satellite has a nadir resolution of $24 \times$

115 $13$ km$^2$ and overpasses the equator around 13:45 LT. More detailed information about these

116 instruments is summarized in Table S1. These instruments measure transmitted, backscattered,

117 and reflected radiation from the atmosphere in the ultraviolet and visible wavelength. The

118 radiation measurements in the wavelength of 402 - 465 nm are then used to retrieve $NO_2$ VCDs.

119 The retrieval process consists of three steps: 1) converting radiation observations to $NO_2$ slant

120 column densities (SCDs) by using the Differential Optical Absorption Spectroscopy (DOAS)





spectral fitting method; 2) separating tropospheric SCDs and stratospheric SCDs from the total
$NO_2$ SCDs; 3) dividing the $NO_2$ tropospheric SCDs by the tropospheric air mass factors (AMF) to
compute VCDs.

The product archives we use in this study include GOME-2B (TM4NO2A v2.3),

SCIAMACHY (QA4ECV v1.1), GOME-2A (QA4ECV v1.1), OMI (QA4ECV v1.1, hereafter
referred to as OMI-QA4ECV), OMNO2 (SPv3, hereafter referred to as OMI-NASA), and the
Berkeley High-Resolution $NO_2$ products (v3.0B, hereafter referred to as OMI-BEHR). OMI-
BEHR uses the tropospheric SCDs from OMI-NASA products but updates some inputs for the
tropospheric AMF calculation (Laughner et al., 2018). These product archives have been
previously validated (Boersma et al., 2018; Drosoglou et al., 2017; Drosoglou et al., 2018;
Krotkov et al., 2017; Laughner et al., 2018; Wang et al., 2017; Zara et al., 2018). Generally, the
pixel-size uncertainties of these products are > 30% over polluted regions under clear-sky
conditions. We summarize the basic information about these products in Table S2. To keep the
high quality and sampling consistency of $NO_2$ TVCD datasets, we chose pixel-size $NO_2$ TVCD
data using the criteria listed in Table S3. After the selection, we re-grid the pixel-size data into the
REAM $36 \times 36$ $km^2$ grid cells and calculate the seasonal means of each grid cell with
corresponding daily values on weekdays (winter: January, February, and December; spring:
March, April, and May; summer: June, July, and Autumn; autumn: September, October, and
November). We excluded weekend data in this study to minimize the impacts of weekend $NO_x$
emission reduction, leading to different $NO_2$ TVCDs between weekdays and weekends (Figure
S2).

Satellite TVCD measurements can show large variations and apparent discontinuities due in

part to the effects of cloud, lightning $NO_x$, the shift of satellite pixel coverage, and retrieval
uncertainties (Figure S2; e.g., (Boersma et al., 2018; Zhang et al., 2018)). However, continuous
and consistent measurements are required for reliable trend analyses. In addition to the criteria of





data selection in Table S3, we compute the seasonal relative 90th percentile confidence interval,
defined as RCI = (X(95th percentile) - X(5th percentile)) / mean(X), where X is the daily $NO_2$
TVCD for a given season. To compute the seasonal trend, we require that RCI is < 50% for the
selected season every year in the analysis period (Table S3). About 45% of data are removed as a
result.
**2.3 Surface $NO_2$ measurements**
Hourly surface $NO_2$ measurements from 2003 - 2017 are from the EPA AQS monitoring
network (archived on https://www.epa.gov/outdoor-air-quality-data). Most AQS monitoring sites
use the Federal Reference Method (FRM) — gas-phase chemiluminescence to measure $NO_2$. Few
sites use the Federal Equivalent Method (FEM) – photolytic-chemiluminescence or the Cavity
Attenuated Phase Shift Spectroscopy (CAPS) method. FRM and FEM are indirect methods, in
which $NO_2$ is first converted to NO and then NO is measured through chemiluminescence
measurement of $NO_2^*$ produced by $NO + O_3$. The difference is that FRM uses heated
reducers/catalysts for the conversion of $NO_2$ to NO and FEM uses photolysis of $NO_2$ to NO. The
conversion to NO in the FRM instruments is not specific to $NO_2$, and non-$NO_x$ active nitrogen
compounds ($NO_z$) can also be reduced by the catalysts, which would cause high biases of $NO_2$
measurements, while the FEM method is sensitive to the photolysis conversion efficiency of $NO_2$
to NO (Beaver et al., 2012; Beaver et al., 2013; Lamsal et al., 2015). The CAPS method directly
determines $NO_2$ concentrations based on a $NO_2$-induced phase shift measured by a photodetector.
The CAPS instrument operates at a wavelength of about 450 nm and may overestimate $NO_2$
concentrations due to absorption of other molecules at the same wavelength (Beaver et al., 2012;
Beaver et al., 2013; Kebabian et al., 2005).
Due to the different characteristics of the above three methods and demonstrated biases
between the FRM and the FEM by Lamsal et al. (2015), we firstly investigate the measurement





discrepancies among the above three methods. There are three sites having FRM and FEM
measurements simultaneously during some periods from 2013 - 2014, two sites having both FRM
and CAPS data during some periods from 2015 – 2016, and one site using all three measurement
methods during some periods in 2015. Figure S3 shows the hourly averaged ratios of FEM and
CAPS to FRM data, respectively, for 4 seasons during 2013 – 2016. The CAPS/FRM ratios are in
the range of 0.94 – 1.06 and the FEM/FRM ratios of 0.86 – 1.11. Furthermore, Zhang et al.
(2018) discussed that the relative trends are not affected by scaling the observation data. As in the
work by Zhang et al. (2018), we analyze the relative trends in the surface $NO_2$ data. We,
therefore, did not scale the FRM data. At sites with FEM or CAPS measurements, we use these
measurements in place of FRM data. If both FEM and CAPS data are available, we use the
averages of the two datasets.

Since $NO_2$ surface concentrations have significant diurnal variations (Figure S4), we choose

the data at 9:00-10:00 LT for comparison with GOME-2A/2B data, 10:00-11:00 LT for
comparison with SCIAMACHY data, and 13:00-14:00 LT for OMI data. The seasonal $RCI <$
50% requirement is also used here to be consistent with the analysis of satellite TVCD data. We
also require that the measurement site must have valid measurements in the aforementioned 3
hours for at least one season from 2003 – 2017. The locations of the 179 selected sites using the
site selection criteria are shown in Figure 1. The region definitions follow the U.S. Census Bureau
(https://www2.census.gov/geo/pdfs/maps-data/maps/reference/us_regdiv.pdf).





## 3. Results and Discussions


### 3.1 Nonlinear relationships among anthropogenic $NO_x$ emissions, $NO_2$ surface concentrations, and $NO_2$ TVCDs



$NO_2$ surface concentrations and $NO_2$ TVCD are not linearly correlated with $NO_x$ emissions
due in part to chemical nonlinearity (Gu et al., 2013; Lamsal et al., 2011). Therefore, it is
necessary to first investigate the nonlinearities among $NO_x$ emissions, $NO_2$ surface
concentrations, and TVCDs over the CONUS before we compare the trends between $NO_2$ surface
concentrations and TVCDs. The nonlinearity between $NO_x$ emission and $NO_2$ TVCD is analyzed
by examining the local sensitivity of $NO_2$ TVCD to $NO_x$ emissions (Gu et al., 2013; Lamsal et al.,
2011; Tong et al., 2015), which is defined as $\beta$ in Equation (1). We further define $\gamma$ as the
sensitivity of $NO_2$ surface concentration to $NO_x$ emission:
$$\frac{\Delta E}{E} = \beta \frac{\Delta \Omega}{\Omega} \qquad (1)$$

$$\frac{\Delta E}{E} = \gamma \frac{\Delta c}{c} \qquad (2)$$

where $E$ denotes $NO_x$ emission and $\Delta E$ denotes the change of $NO_x$ emission; $\Omega$ denotes $NO_2$
TVCD, $c$ denotes surface $NO_2$ concentration, and $\Delta \Omega$ and $\Delta c$ denote the corresponding changes.
We computed $\beta$ and $\gamma$ values for July 2011 over the CONUS using REAM. To compute
local $\beta$ and $\gamma$ values, we added another independent group of chemistry species ("group 2") in
REAM in order to compute the standard and sensitivity simulations concurrently. The original
chemical species in the model ("group 1") were used in the standard simulation. For group 2
chemical species, anthropogenic $NO_x$ emissions were reduced by 15%. In model simulation, we
first computed the advection of group 1 tracers. The horizontal tracer fluxes were therefore


available. All influxes into a grid cell for group 2 tracer simulation were from group 1 tracer
simulation; only outfluxes were computed using group 2 tracers. The outflux was one way in that
nitrogen species were transported out but the transport did not affect adjacent grid cells because
the influxes were from group 1 tracer simulation. Using this procedure, the effects of
anthropogenic $NO_x$ emission reduction were localized. The $\beta$ and $\gamma$ values were computed by the
ratio of TVCD and surface concentration changes to 15% change of anthropogenic $NO_x$
emissions, respectively.

Figure 2 shows the distributions of our $\beta$ and $\gamma$ ratios as a function of anthropogenic $NO_x$

emissions for July 2011 over the CONUS. Results essentially the same as Figure 2 were obtained
when a perturbation of 10% was used for anthropogenic $NO_x$ emissions. While the model
simulation is for one summer month, several key points on the surface and column concentration
sensitivities to anthropogenic $NO_x$ emissions have implications for comparing the trends of AQS
and satellite TVCD data. (1) Both $\beta$ and $\gamma$ values are negatively correlated with anthropogenic
$NO_x$ emissions due to chemical nonlinearity and background $NO_x$ contributions (Gu et al., 2016;
Lamsal et al., 2011). It is consistent with the distribution of $\beta$ as a function of $NO_x$ emissions in
China (Gu et al., 2013), although the $\beta$ ratios for the US are generally larger than for China due
primarily to different emission distributions of $NO_x$ and VOCs and regional circulation patterns
(Zhao et al., 2009b). (2) The uncertainties of $\beta$ and $\gamma$ values increase significantly as
anthropogenic $NO_x$ emissions decrease, which means regions with low anthropogenic $NO_x$
emissions are more sensitive to environmental conditions, such as $NO_x$ transport from nearby
regions which may even produce negative $\beta$ and $\gamma$ values. (3) The value of $\gamma$ is generally less than
$\beta$, especially for low-anthropogenic-$NO_x$ emission regions, which reflects the significant
contribution of free tropospheric $NO_2$ to $NO_2$ TVCD but not to $NO_2$ surface concentrations. (4)
The variations of $\beta$ and $\gamma$ values in anthropogenic $NO_x$ emission bins tend to be larger at 10:00 –
11:00 than at 13:00 – 14:00 LT, reflecting a stronger transport effect due to weaker chemical





losses at 10:00 – 11:00. (5) Both $\beta$ and $\gamma$ values are significantly less than 1 at 13:00 – 14:00 LT
($\beta = 0.74$ and $\gamma = 0.84$) when anthropogenic $NO_x$ emissions are $> 4 \times 10^{12}$ molecules $cm^{-2}$ $s^{-1}$, but
they are close to 1 at 10:00 – 11:00 LT ($\beta = 0.96$ and $\gamma = 1.02$), which reflect stronger chemistry
nonlinearity at 13:00 – 14:00 than in the morning.
The largely varying $\beta$ and $\gamma$ values for anthropogenic $NO_x$ emissions $< 10^{11}$ molecules $cm^{-2}$
$s^{-1}$ imply that the trends derived from satellite TVCD data do not directly represent anthropogenic
$NO_x$ emissions and that the variations of TVCD data may not be comparable to the corresponding
surface $NO_2$ concentrations. We define a region "urban" if anthropogenic $NO_x$ emissions are $>$
$10^{11}$ molecules $cm^{-2}$ $s^{-1}$. All the other regions are defined as "rural". Figure 3 shows the
distributions of anthropogenic $NO_x$ emissions and urban and rural regions defined in this study.
Such defined urban regions account for 69.8% of the total anthropogenic $NO_x$ emissions over the
CONUS, the trend of which is, therefore, representative of anthropogenic emission changes. A
caveat is that some "urban" regions would become "rural" if anthropogenic $NO_x$ emissions
decreased after 2011 as the EPA anthropogenic $NO_x$ emission trend suggested (Figure S1). In a
sensitivity study, we define an urban region using a stricter criterion of anthropogenic $NO_x$
emissions $> 2 \times 10^{11}$ molecules $cm^{-2}$ $s^{-1}$ and the analysis results are similar to those shown in the
next section.
**3.2 Trend comparisons between NO₂ AQS surface concentrations and coincident**
**satellite NO₂ tropospheric VCD over urban and rural regions**
By using anthropogenic $NO_x$ emissions of $10^{11}$ molecules $cm^{-2}$ $s^{-1}$ as the threshold value, 157
AQS sites are urban, and the rest 22 sites are rural. Their properties are summarized in Table 2.
Figure 4 shows the relative annual variations of AQS $NO_2$ surface measurements at 13:00 – 14:00
and coincident OMI-QA4ECV $NO_2$ TVCD data from 2005 – 2017 in each season for urban and
rural regions. The contrast between the two regions is apparent in all seasons. For comparison



purposes, we scale the time series of TVCD and AQS surface $NO_2$ to their corresponding 2005
values, and the resulting data are therefore unitless. Over urban regions, $NO_2$ surface
concentrations are highly correlated with $NO_2$ TVCDs (TVCD = 1.03 × AQS + 0.11, $R^2$ = 0.98),
reflecting the comparable and stable β and γ values (Figure 2). However, over rural regions, the
scaled TVCD data significantly deviate from AQS $NO_2$ data (TVCD = 1.15 × AQS + 0.09, $R^2$ =
0.87). It is noteworthy that the discrepancies between urban and rural data are smaller in winter
than in spring, summer, and autumn due to a more dominant role of transport than chemistry and
lower natural $NO_x$ emissions in winter.

We also examine the correlations of AQS $NO_2$ surface concentrations with coincident OMI-

NASA, OMI-BEHR, SCIAMACHY, GOME-2A, and GOME-2B TVCD measurements. The
results of OMI-NASA and OMI-BEHR are similar to those of OMI-QA4ECV (Figure 4).
SCIAMACHY and GOME-2B TVCD observations at 9:00-11:00 LT also show large contrast
between urban (SCIAMACHY: TVCD = 0.92 × AQS - 0.005, $R^2$ = 0.94; GOME-2B: TVCD =
0.54 × AQS + 0.56, $R^2$ = 0.96) and rural regions (SCIAMACHY: TVCD = 0.77 × AQS +0.83, $R^2$
= 0.63; GOME-2B: TVCD = 0.46 × AQS + 0.73, $R^2$ = 0.59). The correlation of coincident
GOME-2A $NO_2$ TVCD data with AQS surface concentrations is poor for rural (TVCD = 0.65 ×
AQS + 0.56, $R^2$ = 0.44) and urban (TVCD = 0.31 × AQS + 0.56, $R^2$ = 0.21) regions (Figure S5),
which likely reflects the degradation of the GOME-2A instrument causing significant increase of
$NO_2$ SCD uncertainties (Boersma et al., 2018). Therefore, we excluded GOME-2A in the analysis
hereafter.

We further investigate the sensitivities of OMI-QA4ECV $NO_2$ TVCD relative annual

variations from 2005 - 2017 to different anthropogenic $NO_x$ emissions over the CONUS in Figure
5. We find clear flattening of $NO_2$ TVCD variations as anthropogenic $NO_x$ emissions decrease,
which is consistent with the above analysis. Similar to Figure 4, the spread of TVCD variation is
much less in winter than the other seasons. The differences between Figures 5 and 4 are due to a





much larger dataset used in the former than the latter. Only coincident AQS and OMI-QA4ECV
data are used in Figure 4, but all OMI-KMNI data are used in Figure 5.

**3.3 Trend analysis of AQS NO$_2$ surface concentrations, satellite TVCDs, and updated EPA NOx emissions**

**3.3 Trend analysis of AQS NO$_2$ surface concentrations, satellite TVCDs, and**
**updated EPA NOx emissions**
We first updated the CEMS measurement data used in the EPA NO$_x$ emission trend datasets
with the newest datasets obtained from https://ampd.epa.gov/ampd/. As shown in Figure S1, the
updated CEMS data lead to a reduction of anthropogenic NO$_x$ emissions during the Great
Recession (2008 – 2009) and a recovery period in 2010 – 2011. The sharp drop during the Great
Recession and the flattening trend right after the Great Recession are captured by OMI NO$_2$ and
SCIAMACHY TVCD products (Figures 4, 6, and S6) and AQS NO$_2$ surface measurements
(Figures 4, 6, and S4) and are also noted by Russell et al. (2012) and Tong et al. (2015) (Table 1).
In Figure 6, we show the comparisons among the relative variations of the updated EPA
anthropogenic NO$_x$ emissions, AQS NO$_2$ surface measurements at 10:00-11:00 and 13:00-14:00,
and coincident satellite NO$_2$ TVCDs for urban regions in 4 seasons from 2003 to 2017. Also
shown are the comparisons among the updated EPA anthropogenic NO$_x$ emissions and satellite
NO$_2$ TVCDs. There are many more data points for the latter comparison because the data
selection is no longer limited to those coincident with the AQS surface data, and therefore, the
uncertainty spread is much lower. The comparisons, in general, show consistent results that the
updated EPA anthropogenic NO$_x$ emissions, AQS surface measurements, and satellite TVCD
data are in agreement. The agreement of decreasing trends among the datasets is just as good for
the post-2011 period as the pre-2011 period. This result differs from Miyazaki et al. (2017) and
Jiang et al. (2018), who suggested no significant decreasing trend for OMI TVCD data and
inversed NO$_x$ emissions after 2010. The disagreement can be explained by the results of Figure 5.
Including the low anthropogenic NO$_x$ emission regions leads to underestimates of NO$_x$ decreases.



Since the area of low anthropogenic $NO_x$ emission regions is larger than high anthropogenic $NO_x$
emission regions (Table 2), the arithmetic averaging will lead to a large weighting of rural
observations, which do not reflect anthropogenic $NO_x$ emission changes. Miyazaki et al. (2017)
and Jiang et al. (2018) included all regions in their analyses, but we exclude rural regions. Figure
S6 shows the seasonal variations if the TVCDs over rural regions are included; the result shows a
much lower decreasing rate of TVCDs over the CONUS. The much slower satellite TVCD trends
for regions with low $NO_x$ emissions was previously discussed by Zhang et al. (2018). In addition,
Miyazaki et al. (2017) and Jiang et al. (2018) conducted $NO_x$ emission inversions by using the
Model for Interdisciplinary Research on Climate (MIROC)-Chem with a coarse resolution of 2.8°
× 2.8°, which was insufficient to separate urban and rural regions and might distort predicted $NO_2$
TVCDs and inversed $NO_x$ emissions due to nonlinear effects (Valin et al., 2011; Yu et al., 2016),
which is another possible reason for their find of flattening $NO_x$ emission trends after 2010.

We summarize the decreasing rates of $NO_2$ after the Great Recession in Table 3. To

minimize the effect of the sharp decrease and the subsequent recovery, we chose to analyze the
post-2011 period. Table 3 summarizes the results for each season, while Table 1 gives the
averaged annual decreasing trends. Generally, Tables 1 and 3 confirm the continuous decreases of
AQS surface observations, satellite $NO_2$ TVCD, and updated EPA anthropogenic $NO_x$ emissions
after 2011 as in Figure 6, but decreasing rates are lower than the pre-2011 period. Over the AQS
urban sites, the slowdown magnitudes are 9% for AQS surface observations and 20% - 40% for
satellite $NO_2$ TVCD measurements, which may reflect in part smaller γ than β values (Table 2).
Our estimated slowdown magnitudes are significantly lower than Lamsal et al. (2015) and Jiang
et al. (2018) but comparable to the results by Tong et al. (2015) (Table 1). The agreement with
Tong et al. (2015) is because we select urban AQS sites based on anthropogenic $NO_x$ emissions
and they chose eight large cities, while Lamsal et al. (2015) and Jiang et al. (2018) used all AQS
sites.





Over the CONUS urban regions, updated EPA anthropogenic $NO_x$ emissions show a
slowdown of 22% compared to 29% - 46% for three OMI $NO_2$ TVCD products. The difference is
partially due to the β ratio of $2.3 \pm 0.9$ at 13:00 – 14:00 over the CONUS urban regions (Table 2).
Satellite $NO_2$ TVCD measurement uncertainties also contribute to the difference. From 2013 –
2017, GOME-2B $NO_2$ TVCDs decrease more than OMI products, especially in spring, autumn
and winter (Tables 1 and 3). Finally, trend analyses in different regions (Figure 7 and Table S4)
indicate that generally, the Midwest has the least slowdown of the decreasing rate for urban OMI
$NO_2$ TVCD (-14% on average) after 2011 compared to the Northeast (-30%), South (-34%), and
West (-28%).
The results presented in this study are qualitatively in agreement with the work by Silvern et
al. (2019). The two studies were independent. Therefore, the foci of the studies are different
despite reaching similar conclusions. While we focused on understanding the detailed data
analysis of Jiang et al. (2018) and limited the use of model simulation results so that our results
can be compared to the previous study directly, Silvern et al. (2019) relied more on multi-year
model simulations. As a result, Silvern et al. (2019) can clearly identify the contributions of the
$NO_2$ columns by natural emissions and make use of additional observations such as nitrate
deposition fluxes. They also identified model biases in simulating the trends of $NO_2$ TVCDs by
natural emissions. Our study, on the other hand, explored the data analysis procedure through
which the trend of anthropogenic emissions can be derived from satellite observations and its
limitations.

## 4. Conclusions

Using model simulations for July 2017, we demonstrate the nonlinear relationship of $NO_2$
surface concentration and TVCD with anthropogenic $NO_x$ emissions. Over low anthropogenic
$NO_x$ emission regions, the ratios of anthropogenic $NO_x$ emission changes to the changes of



surface concentrations (γ) and TVCDs (β) have very large variations and $\beta > \gamma \gg 1$.
Therefore, for the same emission changes, surface concentration and TVCD changes are much
smaller and variable than urban regions, making it difficult to use the observations to directly
infer anthropogenic $NO_x$ emission trends. We find that defining urban regions where
anthropogenic $NO_x$ emissions are $> 10^{11}$ molecules $cm^{-2}$ $s^{-1}$ and using surface and TVCD
observations over these regions can infer the trends that can be compared with the EPA emission
trend estimates.

We evaluate the anthropogenic $NO_x$ emission variations from 2003 – 2017 over the CONUS

by using satellite $NO_2$ TVCD products from GOME-2B, SCIAMACHY, OMI-QA4ECV, OMI-
NASA, and OMI-BEHR, over the urban regions of CONUS. We find broad agreements among
the decreases of AQS $NO_2$ surface observations, satellite $NO_2$ TVCD products, and the EPA
anthropogenic $NO_x$ emissions with the CEMS dataset updated. After 2011, they all show a
slowdown of the decreasing rates. Over the AQS urban sites, $NO_2$ surface concentrations have a
slowdown of 9% and OMI products show a slowdown of 20% - 40%. Over the CONUS urban
regions, OMI TVCD products indicate a slowdown of 29% - 46%, and the updated EPA
anthropogenic $NO_x$ emissions have a slowdown of 22%. The different slowdown magnitudes
between OMI TVCD products and the other two datasets may be caused by the nonlinear
response of TVCD to anthropogenic emissions and the uncertainties of satellite measurements
(e.g., GOME-2B TVCD data show a larger decreasing trend than OMI products from 2013 –

2017).

We did not find observation evidence supporting the notion that anthropogenic $NO_x$

emissions have not been decreasing after the Great Recession. In future studies, we recommend
that the nonlinear relationships of $NO_x$ emissions with $NO_2$ TVCD and surface concentration be





carefully evaluated when applying satellite and surface measurements to infer the changes of
anthropogenic $NO_x$ emissions.

**Data availability**

The EPA AQS hourly surface $NO_2$ measurements are downloaded from
https://aqs.epa.gov/aqsweb/airdata/download_files.html#Raw. QA4ECV 1.1 $NO_2$ VCD products
(OMI-QA4ECV, GOME-2A, and SCIAMACHY) are from http://temis.nl/qa4ecv/no2col/data/.
GOME-2B $NO_2$ VCD products are from
http://www.temis.nl/airpollution/no2col/no2colgome2b.php. OMI-BEHR and OMI-NASA
archives are from http://behr.cchem.berkeley.edu/DownloadBEHRData.aspx. REAM simulation
results for this study are available upon request.

**Author contribution**

JL and YW designed the study. JL conducted model simulations and data analyses with
discussions with YW. JL and YW wrote the manuscript.

**Competing interests**

The authors declare that they have no conflict of interest.

**Acknowledgments**

This work was supported by the NASA ACMAP Program. We thank Ruixiong Zhang for
discussions with J. Li. Thank Benjamin Wells, Alison Eyth, Lee Tooly from EPA, the EPA
MOVES team, Betty Carter from COORDINATING RESEARCH COUNCIL, INC., Brain
McDonald from NOAA, and Zhe Jiang from University of Science and Technology of China for
helping us an understanding of the NEI MOVES mobile source emissions.



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





**Table 1.** Summary of trends of satellite NO₂ TVCD products, NO₂ surface measurements, and EPA anthropogenic NOₓ emissions during from different studies

| Studies | Datasets | Period 1[1] Time | Trend (yr⁻¹)[2] | Period 2 Time | Trend (yr⁻¹) | Period 3 Time | Trend (yr⁻¹) | Slowdown ratio[3] |
|---|---|---|---|---|---|---|---|---|
| This study for CONUS "urban" sites[4] | GOME-2B[5] (36 × 36 km²) | | | | | 2013 - 2017 | -8.2 ± 3.0% | |
| | SCIAMACHY (36 × 36 km²) | 2003 – 2011 | -6.3 ± 1.1% | | | | | |
| | OMI-NASA (36 × 36 km²) | 2005 – 2011 | -8.6 ± 1.2% | | | 2011 – 2016 | -6.1 ± 3.6% | -29%[2] |
| | OMI-BEHR (36 × 36 km²) | 2005 – 2011 | -8.2 ± 1.3% | | | 2011 – 2016 | -4.4 ± 1.6% | -46% |
| | OMI-QA4ECV (36 × 36 km²) | 2005 – 2011 | -7.7 ± 1.4% | | | 2011 - 2017 | -4.2 ± 0.5% | -46% |
| | Updated EPA NOₓ emissions[6] | 2003 – 2011 | -6.5 ± 0.8% | | | 2011 - 2017 | -5.1 ± 0.3% | -22% |
| This study for AQS "urban" sites | GOME-2B (36 × 36 km²) | | | | | 2013 - 2017 | -10.2 ± 2.9% | |
| | SCIAMACHY (36 × 36 km²) | 2003 - 2011 | -7.6 ± 1.1% | | | | | |
| | OMI-NASA (36 × 36 km²) | 2005 - 2011 | -9.0 ± 0.8% | | | 2011 – 2016 | -7.2 ± 3.8% | -20% |
| | OMI-BEHR (36 × 36 km²) | 2005 - 2011 | -8.9 ± 0.3% | | | 2011 – 2016 | -6.2 ± 2.6% | -30% |
| | OMI-QA4ECV (36 × 36 km²) | 2005 - 2011 | -9.0 ± 0.8% | | | 2011 - 2017 | -5.4 ± 0.9% | -40% |
| | NO₂ surface VMR[7] | 2003 - 2011 | -6.5 ± 1.2% | | | 2011 - 2017 | -5.9 ± 0.8% | -9% |
| (Russell et al., 2012)[8] | BEHR v2.1 NO₂ TVCD (0.05°×0.05°)[9] | 2005 - 2007 | -6 ± 5% (-6.2%)[9] | 2007 - 2009 | -8 ± 5% (-8.4%) | 2009 - 2011 | -3 ± 4% (-3.0%) | -52% |
| | Updated EPA NOₓ emissions | | -6.0% | | -10.0% | | -2.4% | -60% |
| (Tong et al., 2015)[10] | NASA v2.1 NO₂ TVCD (pixels <50 × 24 km²) | 2005 - 2007 | -7.3% (-7.6%) | 2008 - 2009 | -9.2% (-11.4%) | 2010 - 2012 | -2.8% (-4.4%) | -42% |
| | BEHR v2.1 NO₂ TVCD (pixels <50 × 24 km²) | | -8.9% (-9.3%) | | -9.1% (-11.8%) | | -3.6% (-6.0%) | -35% |
| | NO₂ surface VMR | | -6.0% (-6.2%) | | -10.8% (-13.2%) | | -3.4% (-5.4%) | -13% |
| | Updated EPA NOₓ emissions | | -6.0% | | -10.0% | | -3.4% | -43% |
| (Lamsal et al., 2015)[11] | NASA v2.1 NO₂ TVCD (0.1°×0.1°) | 2005 - 2008 | -4.8 ± 1.9% (-5.1%) | | | 2010 - 2013 | -1.2 ± 1.2% (-1.2%) | -76% |
| | NO₂ surface VMR | | -3.7 ± 1.5% (-3.8%) | | | | -2.1 ± 1.4% (-2.1%) | -45% |
| | Updated EPA NOₓ emissions | | -6.4% | | | | -4.0% | -38% |
| (Jiang et al., 2018)[11] | NASA v3 NO₂ TVCD (0.5°×0.667°) | 2005 - 2009 | -10.2 ± 1.8% (-9.8%) | | | 2011-2015 | -3.2 ± 1.6% (-3.2%) | -67% |
| | QA4ECV v2 NO₂ TVCD (0.5°×0.667°) | | -9.6 ± 1.7% (-9.3%) | | | | -2.6 ± 1.8% (-2.6%) | -72% |
| | BEHR v2.1 NO₂ TVCD (0.5°×0.667°) | | -8.5 ± 1.8% (-8.2%) | | | | -2.1 ± 1.6% (-2.1%) | -74% |
| | NO₂ surface VMR | | -6.6 ± 1.4% (-6.4%) | | | | -2.6 ± 1.5% (-2.6%) | -59% |
| | Updated EPA NOₓ emissions | | -7.8% | | | | -5.0% | -36% |

[1] Since different studies used different time division methods, we list the period of each study in the table.
[2] Trends are based on an exponential model $(E(y)) = E_0 \times r^{y-y_0}$; "$y$" denotes year and "$y_0$" denotes the initial year; "$E(y)$" denotes the value at year "$y$" and "$E_0$" denotes the value at the initial year; $r-1$ is the relative trend).
[3] Slowdown ratios = Trend in "period 3" / Trend in "period 1" − 1.
[4] Trends in our study are calculated based on the national seasonal trends shown in Table 3.
[5] The information on satellite products used in this study is summarized in Table S2.
[6] We updated EPA anthropogenic NOₓ emissions with the newest Continuous Emission Monitoring Systems (CEMS) datasets. Figure S1 shows the comparison between our updated and original EPA anthropogenic NOₓ emissions (EPA, 2018).
[7] Denote the averaged trends of 13:00 and 10:00 LT based on the values in Table 3.




[8] The study used $NO_2$ TVCD from urban and power plant grid cells across the U.S.
[9] Since previous studies used linear models to calculate trends and the results are sensitive to their calculation methods and the selection of initial years, we recalculate the trends based on the above exponential model, which makes all the results
consistent. Our results are those bold numbers inside the parentheses, while the numbers in normal fonts are from the original publications.
[10] The study uses $NO_2$ TVCD and surface concentrations from Los Angeles, Dallas, Houston, Atlanta, Philadelphia, Washington, D.C., New York, and Boston.
[11] The two studies used the EPA Air Quality System (AQS) $NO_2$ surface measurements and coincident satellite $NO_2$ TVCD data over the U.S.





**Table 2.** Properties of urban and rural regions in July 2011

| type | Surface area fraction[1] | Anthropogenic $NO_x$ emissions ($\times 10^{10}$ molecules cm$^{-2}$ s$^{-1}$) | $\beta$ at 13:00 – 14:00 LT | $\gamma$ at 13:00 – 14:00 LT | $\beta$ at 10:00 – 11:00 LT | $\gamma$ at 10:00 – 11:00 LT |
|---|---|---|---|---|---|---|
| Urban/CONUS[2] | 17.3% | 29.9 | $2.3 \pm 0.9$ | $1.4 \pm 0.3$ | $2.4 \pm 1.8$ | $1.5 \pm 1.0$ |
| Rural/CONUS | 82.7% | 2.7 | $8.1 \pm 8.7$ | $3.1 \pm 3.9$ | $5.9 \pm 8.0$ | $2.8 \pm 5.8$ |
| Urban/AQS | 87.7% | 71.0 | $1.5 \pm 0.7$ | $1.2 \pm 0.4$ | $1.7 \pm 1.0$ | $1.3 \pm 0.5$ |
| Rural/AQS | 12.3% | 5.7 | $5.0 \pm 2.0$ | $2.5 \pm 1.3$ | $4.3 \pm 3.2$ | $2.7 \pm 2.6$ |

[1] "Fraction" denotes the percentages of "urban" or "rural" data points for the whole CONUS or all AQS sites.
[2] "Urban-CONUS" denote CONUS "urban" grid cells; "Urban-AQS" denote AQS "urban" site grid cells.





**Table 3.** Summary of national trends of updated EPA anthropogenic $NO_x$ emissions, AQS $NO_2$ surface concentrations at 13:00 − 14:00 and 10:00 − 11:00 LT, and satellite $NO_2$ TVCD products for 4 seasons during different periods[1]

| | | Spring | | Summer | | Autumn | | Winter | |
|---|---|---|---|---|---|---|---|---|---|
| | | AQS site | CONUS | AQS site | CONUS | AQS site | CONUS | AQS site | CONUS |
| AQS $NO_2$ VMR at 13:00 -14:00 | 2003 − 2011 | -7.3 ± 1.4% | | -7.4 ± 0.9% | | -6.7 ± 1.8% | | -5.2 ± 0.8% | |
| | 2011 − 2017 | -5.3 ± 1.6% | | -6.4 ± 1.2% | | -7.3 ± 2.5% | | -6.0 ± 2.8% | |
| AQS $NO_2$ VMR at 10:00 − 11:00 | 2003 − 2011 | -7.1 ± 1.6% | | -7.6 ± 1.5% | | -6.2 ± 2.2% | | -4.4 ± 1.6% | |
| | 2011 − 2017 | -4.4 ± 1.4% | | -6.1 ± 1.8% | | -6.3 ± 2.5% | | -5.2 ± 2.4% | |
| SCIAMACHY | 2003 − 2011 | -8.8 ± 3.4% | -6.9 ± 1.1% | -8.2 ± 1.6% | -5.2 ± 1.2% | -6.8 ± 2.4% | -5.6 ± 2.1% | -6.4 ± 7.4% | -7.5 ± 5.5% |
| | 2011 − 2017 | | | | | | | | |
| GOME2B | 2003 − 2011 | | | | | | | | |
| | 2013 − 2017 | -10.2 ± 7.8% | -8.3 ± 16.9% | -6.4 ± 14.0% | -5.3 ± 4.0% | -10.5 ± 41.6% | -6.9 ± 13.2% | -13.6 ± 15.1% | -12.3 ± 78.9% |
| OMI-QA4ECV | 2005 − 2011 | -9.3 ± 5.6% | -8.3 ± 4.6% | -8.3 ± 2.4% | -5.9 ± 5.2% | -10.0 ± 4.2% | -7.4 ± 2.4% | -8.3 ± 2.1% | -9.3 ± 5.2% |
| | 2011 − 2017 | -5.3 ± 6.0% | -4.3 ± 6.5% | -4.2 ± 3.0% | -4.9 ± 9.2% | -6.0 ± 1.8% | -3.8 ± 1.8% | -6.1 ± 25.6% | -3.8 ± 3.5% |
| OMI-NASA | 2005 − 2011 | -9.4 ± 5.0% | -9.6 ± 5.3% | -9.4 ± 2.8% | -7.1 ± 2.9% | -9.4 ± 3.2% | -8.1 ± 2.8% | -7.8 ± 3.6% | -9.5 ± 16.6% |
| | 2011 − 2016 | -4.4 ± 18.9% | -3.8 ± 7.5% | -5.7 ± 6.7% | -4.5 ± 5.3% | -6.0 ± 3.1% | -4.6 ± 3.9% | -12.8 ± 7.8% | -11.4 ± 6.6% |
| OMI-BEHR | 2005 − 2011 | -9.1 ± 5.3% | -8.9 ± 5.8% | -8.7 ± 2.4% | -6.4 ± 3.2% | -9.2 ± 3.2% | -8.0 ± 3.1% | -8.5 ± 10.6% | -9.4 ± 23.0% |
| | 2011 − 2016 | -3.8 ± 4.4% | -3.0 ± 4.0% | -5.4 ± 7.0% | -3.9 ± 6.6% | -5.6 ± 13.2% | -4.1 ± 14.0% | -9.9 ± 5.2% | -6.7 ± 5.9% |
| EPA | 2003 − 2011 | | | -6.5 ± 0.8% | | | | | |
| | 2011 − 2017 | | | -5.1 ± 0.3% | | | | | |

[1] We calculate trends by using the exponential model described in Table 1.





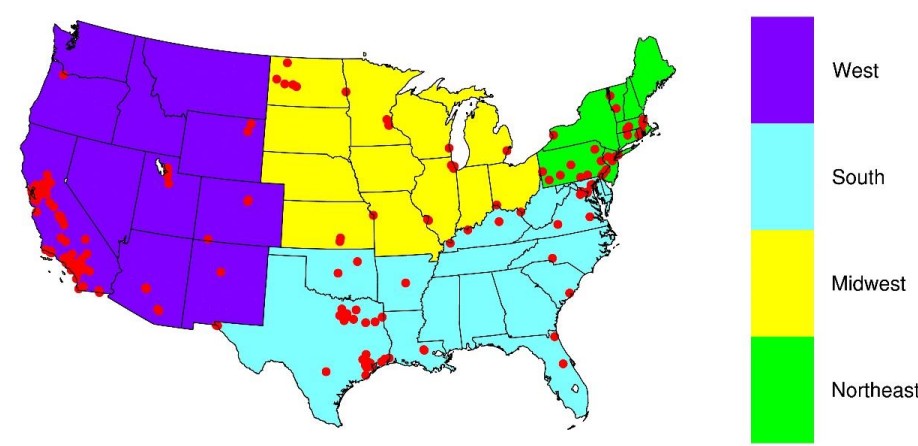

Figure 1. Region definitions and locations of $NO_2$ surface observation sites used in this study.






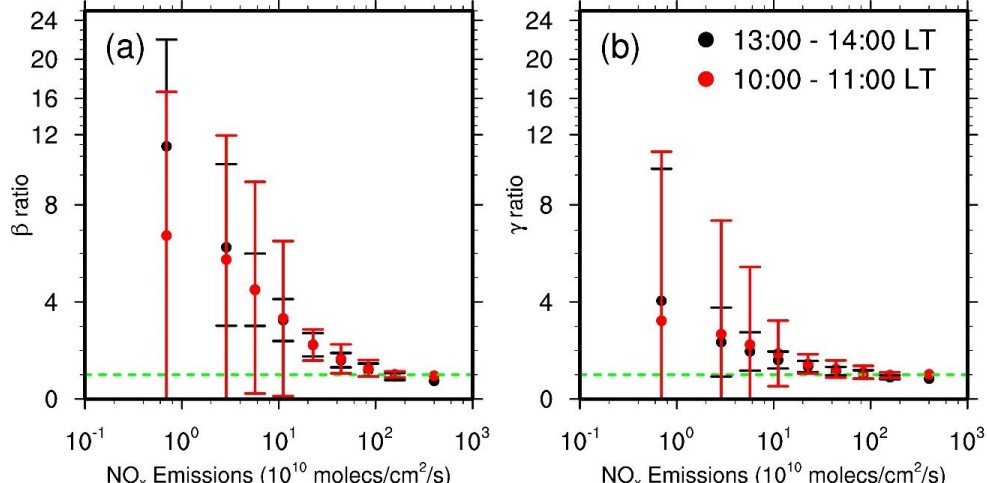

Figure 2. Distributions of β (panel a) and γ (panel b) ratios as a function of anthropogenic $NO_x$
emissions on weekdays for July 2011 over the CONUS. "13:00 – 14:00 LT" is for OMI, and
"10:00 – 11:00" LT is for SCIAMACHY and GOME-2A/2B. The data are binned into nine
groups based on anthropogenic $NO_x$ emissions: $E \in (0, 2^1), [2^1, 2^2), [2^2, 2^3), [2^3, 2^4), [2^4, 2^5), [2^5,$
$2^6), [2^6, 2^7), [2^7, 2^8), [2^8, 2^9) \times 10^{10}$ molecules $cm^{-2} s^{-1}$. Here, $(0, 2^1)$ denotes $0 <$ emissions $< 2^1$,
and $[2^1, 2^2)$ denotes $2^1 \leq$ emissions $< 2^2$, similar to other intervals. The green dashed line denotes
a value of 1. Error bars denote standard deviations.

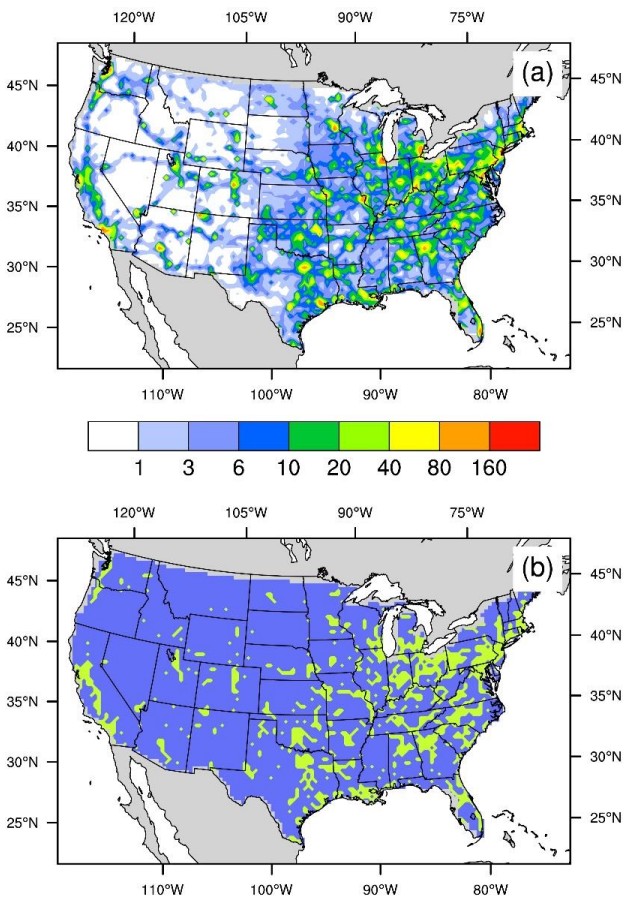

Figure 3. Spatial distributions of (a) anthropogenic NO$_x$ emissions (unit: $10^{10}$ molecules cm$^{-2}$ s$^{-1}$)
and (b) "urban" regions satisfying our selection criteria. In (b), light green and blue denote the
resulting urban and rural regions, respectively.




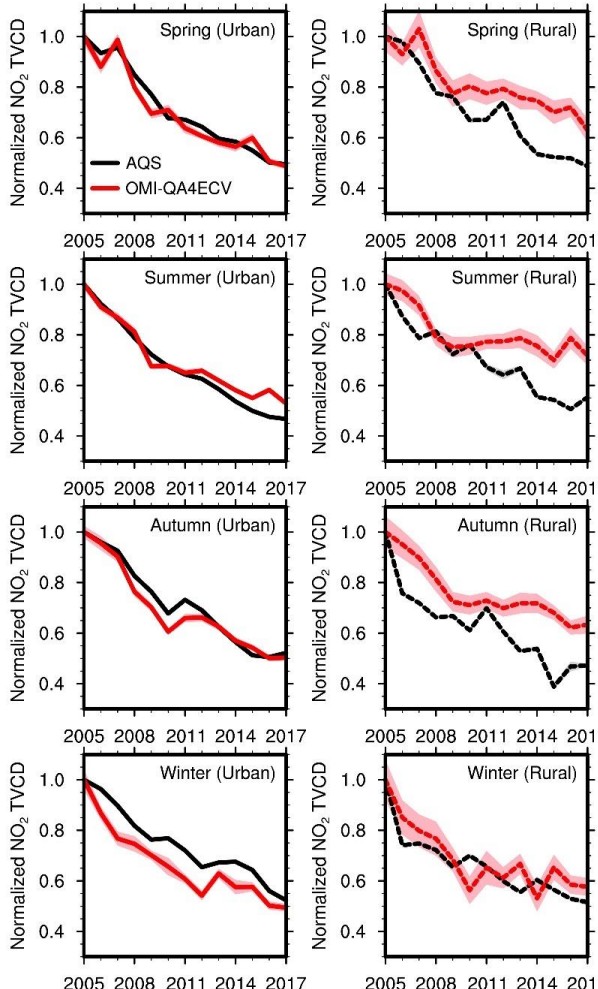

Figure 4. Relative annual variations of AQS NO$_2$ surface concentrations and coincident OMI-
QA4ECV NO$_2$ TVCD in each season from 2005 – 2017 for urban (left panel) and rural (right
panel) regions. The observation data are scaled by the corresponding 2005 values. Black and red
lines denote AQS surface observations and OMI-QA4ECV NO$_2$ TVCDs, respectively. Shading in
a lighter color is added to show the standard deviation of the results; when uncertainty is small
due in part to a large number of data points, shading area may not show up.



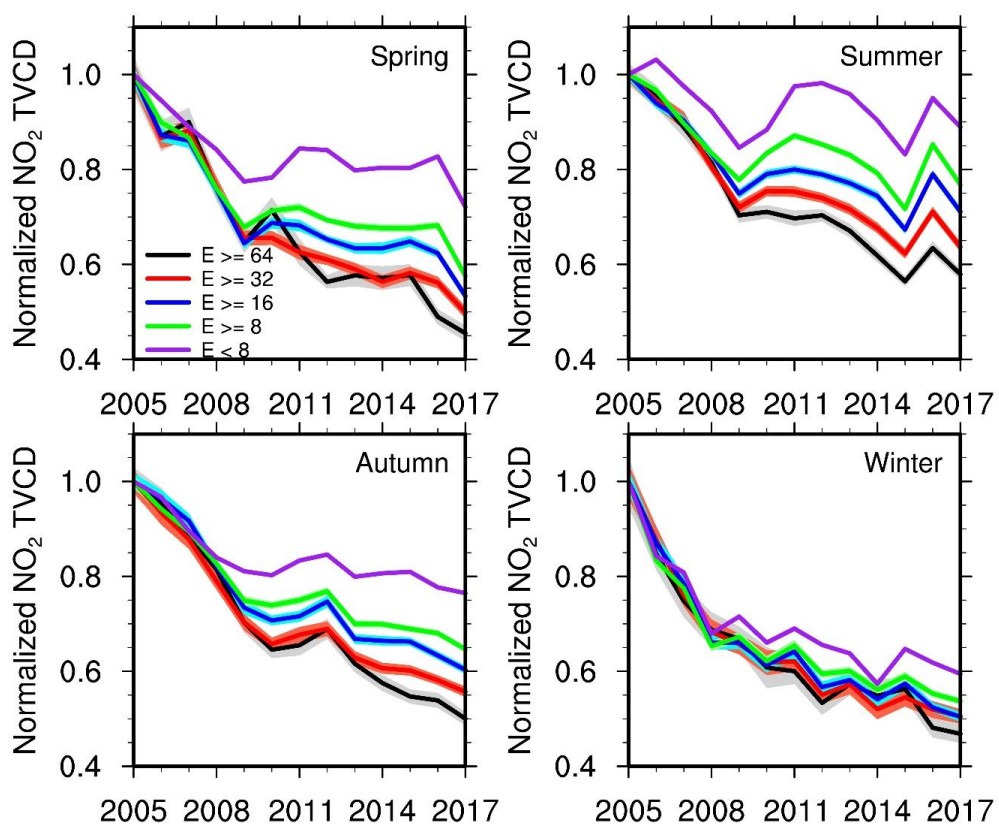


Figure 5. Relative annual variations of OMI-QA4ECV $NO_2$ TVCD for different anthropogenic
$NO_x$-emission groups in each season from 2005 – 2017. "E >= 64" denotes grid cells with
anthropogenic $NO_x$ emissions over $64 \times 10^{10}$ molecules $cm^{-2}$ $s^{-1}$. "E >= 32" denotes grid cells
with anthropogenic $NO_x$ emissions equal to or larger than $32 \times 10^{10}$ molecules $cm^{-2}$ $s^{-1}$ but less
than $64 \times 10^{10}$ molecules $cm^{-2}$ $s^{-1}$. "E >= 16" and "E >= 8" have similar meanings as "E >= 32".
"E < 8" denotes grid cells with anthropogenic $NO_x$ emissions less than $8 \times 10^{10}$ molecules $cm^{-2}$ $s^{-}$
$^{1}$. Shading in a lighter color is added to show the standard deviation of the results; when
uncertainty is small due in part to a large number of data points, shading area may not show up.


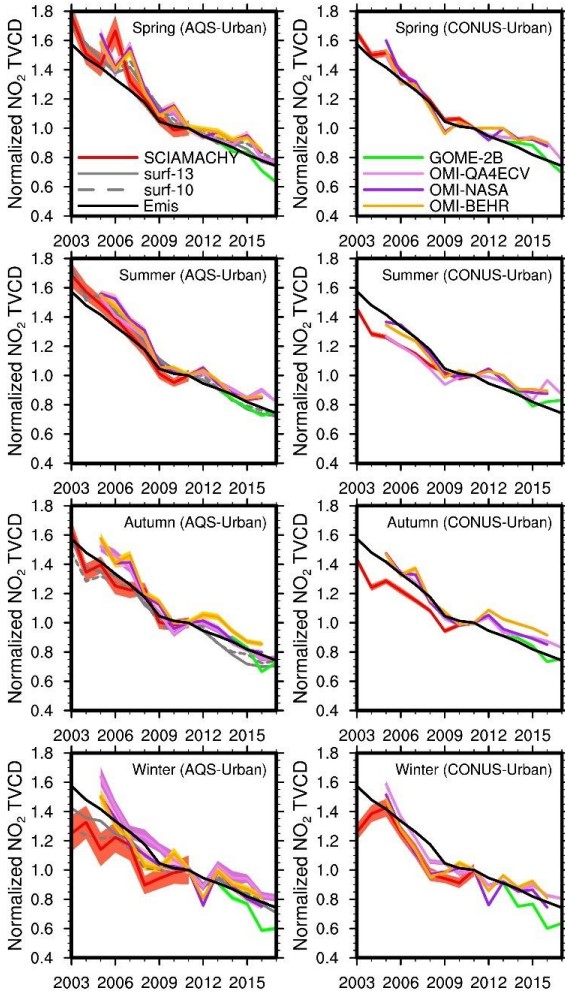

Figure 6. Relative variations of AQS NO₂ surface measurements at 13:00-14:00 and 10:00-11:00
LT, updated EPA anthropogenic NOₓ emissions, and satellite NO₂ TVCD data over the AQS
urban sites (left column) and the CONUS urban regions (right column) for 4 seasons. AQS NO₂
surface measurements are not included in the right column. All datasets are scaled by their
corresponding values in 2011 except for GOME-2B. For GOME-2B, we firstly normalized the
values in each season to the corresponding 2013 values and plotted the relative changes from the
2013 EPA point of each season to make the GOME-2B relative variations comparable to the
other datasets. Shading in a lighter color is added to show the standard deviation of the results;
when uncertainty is small due in part to a large number of data points, shading area may not show
up.




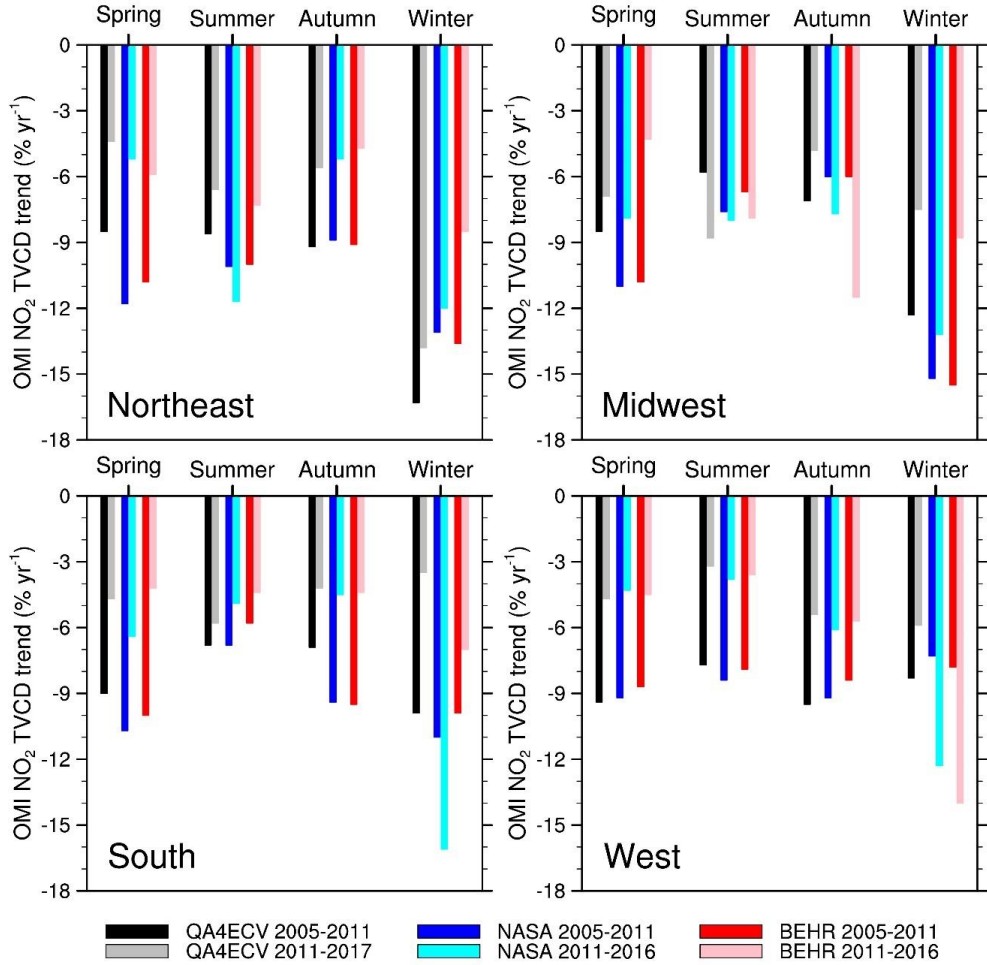


Figure 7. Pre- and post-2011 OMI NO$_2$ TVCD trends for 4 seasons in the urban regions of
Northeast, Midwest, South, and West. Black bars denote OMI-QA4ECV NO$_2$ TVCD trends from
2005 – 2011; gray bars denote the corresponding trends during 2011 – 2017. Blue bars denote
OMI-NASA trends from 2005 – 2011; cyan bars denote NASA-OMI trends from 2011 – 2016.
Red bars denote BEHR-OMI trends from 2005 – 2011; pink bars denote OMI-BEHR trends from
2011 – 2016.

700