# Peer review of "Inferring the anthropogenic NOₓ emission trend over the United States during 2003 - 2017 from satellite observations: Was there a flattening of the emission tend after the Great Recession?"

_Atmospheric Chemistry and Physics, 2019_

## Short Comment (SC1) · 16 Jul 2019

Dear authors, in support of your results I would like to bring your attention to a recent study on satellite-based tropospheric NO2 trends and trend reversals (1996-2017). In this study, it is shown that several regions in the US experienced a trend reversal around the period 2000 from positive or neutral trends to negative ones. There are also results for selected megacities in the US (e.g. Los Angeles).

[Figure]

Georgoulias, A. K., van der A, R. J., Stammes, P., Boersma, K. F., and Eskes, H. J.: Trends and trend reversal detection in 2 decades of tropospheric NO2 satellite observations, Atmos. Chem. Phys., 19, 6269-6294, https://doi.org/10.5194/acp-19-6269-2019, 2019.

---

## Author Comment (AC1) · 28 Jul 2019

**Response to Aristeidis Georgoulias**

Thank you for your useful suggestions. Our answers follow your comments (in *Italics*).

*Comments/suggestions:*

*Dear authors, in support of your results I would like to bring your attention to a recent study on satellite-based tropospheric NO2 trends and trend reversals (1996-2017). In this study, it is shown that several regions in the US experienced a trend reversal around the period 2000 from positive or neutral trends to negative ones. There are also results for selected megacities in the US (e.g. Los Angeles).*

*Georgoulias, A. K., van der A, R. J., Stammes, P., Boersma, K. F., and Eskes, H. J.: Trends and trend reversal detection in 2 decades of tropospheric NO2 satellite observations, Atmos. Chem. Phys., 19, 6269-6294, https://doi.org/10.5194/acp-19-6269-2019, 2019.*

**Reply:**

Thank you for providing a useful reference. Georgoulias et al. (2019) investigated the trends of mid-morning $NO_2$ tropospheric vertical column densities around the world at multi-spatial scales from $1996 - 2017$ based on GOME, SCIAMACHY, GOME-2A, and GOME-2B products, confirming significant decreases of $NO_2$ TVCDs over the United States in the recent two decades. The paper is quite relevant to our work, and we will cite the paper during the revision of the manuscript.

**References**

Georgoulias, A. K., van der A, R. J., Stammes, P., Boersma, K. F., and Eskes, H. J.: Trends and trend reversal detection in 2 decades of tropospheric $NO_2$ satellite observations, Atmos. Chem. Phys., 19, 6269-6294, 10.5194/acp-19-6269-2019, 2019.

---

## Referee Comment (RC1) · Anonymous Referee #1 · 13 Aug 2019

There are a number of recent papers on the topic of NOx emission trends in the United States as observed from space and as compared to predictions from models. The papers raise issues about the emission models, about the resolution of measurements and models needed to derive accurate trends, about interpretation of satellite observations including whether and how the regional background is included in the trend analysis, and whether the lifetime of NOx is also changing with time affecting interpretation of temporal trends. The analysis in this paper focuses on nonlinearities in

chemistry which is related to the question of chemical lifetime. The analysis in the paper seems solid and the discussion and conclusions try to put the paper in context of the recent literature.

I recommend the abstract be revisited in light of the discussion and conclusions as now written.

I also recommend the authors consider whether they can make some more general conclusions about the role of nonlinearities that are the focus of their work as a guide to future research. For example does this research help push forward the conversation about the model resolution needed to describe NOx to a specified accuracy? Other papers suggest that 36km might not be sufficient for the absolute accuracy the authors are trying to achieve. On the other hand there might be cancellation of errors in computation of trends that allows use of lower resolution for questions about trends?

---

## Referee Comment (RC2) · Anonymous Referee #2 · 16 Aug 2019

There has been many studies in the last years on the recent trends of NOx emissions over the U.S., the main motivation being the apparent important change in NO2 column trend since 2010, which obviously requires careful analysis using the available data as well as using models. The present study is useful, as it clearly shows that there is no significant discrepancy between the NEI emission trends and the different NO2 (surface and column) data, when considering only urban areas. The paper discusses the non-linear relationship between NOx emissions and NO2 abundances.

[Figure]

Model calculations using REAM at 36kmx36 km are used to illustrate this point and show that the feedbacks are much stronger at low-NOx than at high-NOx. Although the relevance of NOx natural emissions (which obviously do not have the same trends as the anthropogenic component) is mentioned, the paper does not dwell on it.

In fact, and this is my main comment, I think clarifications are needed in order to sort out the respective roles of chemical non-linearities and the existence of the background. Both natural emissions and chemical non-linearities play their largest role during summer over rural areas, and more so in the free troposphere than near the surface. But it is not entirely clear from the paper how much these two main factors contributed to the apparent discrepancy between the different sets of trends. This should be clarified. Also, although the paper mentions the use of observed $NO_3^-$ deposition trends to further support the declining trend of NOx emissions, it would be useful to incorporate more explicitly this information in the discussion.

Additional (minor) comments:

- l. 34, the total of 0.24 Tg N for natural NOx emissions seems to be very low, where does it come from? I don't think NEI2014 provides this information. Please provide separately the soil, biomass burning and lightning emission information.

- l. 64-65: there are earlier references for the effect of non-linearities on NO2 trends

- section 2.1 on REAM. What is the model domain?

- l. 96: How is meteorology constrained by NCEP?

- l. 100-102 it's a detail, but it seems a little strange that weekday emissions are based on NEI while weekend values are reduced. Isn't NEI an average?

- l. 105 what about lightning emissions?

- l. 148-149 the requirement that RCI > 50% is quite strict. What happens to the trends when you change that?

- l. 184 how many measurements are rejected from this conditions on RCI?

- l. 202 Are the $\beta$ and $\gamma$ calculated based on total emissions with or without lightning emissions? Lightning contributes significantly to the total column, but very little to surface concentrations (in part due to the vertical dependence of spaceborne instruments sensitivity).

- l. 229 "such as NOx transport from nearby regions" this is surprising since the calculated sensitivities were said to be purely local

l. 234 there is no "transport effect". $\beta$ and $\gamma$ are closer to 1 at 10-11 LT (compred to 13-14 LT) because of the weaker chemical losses.

l. 242 I suppose the "urban" definition depends on anthropogenic NOx emissions on a specific year (and month maybe). This should be specified.

l. 330-332 Note that only 22 AQS sites (out of 179) are rural. Therefore, is the difference between this study and the results of Lamsal et al. and Jiang et al. really due to the selection of urban sites?

l. 349-350 the sentence "They also identified model biases (...) natural emissions" is unclear, please either elaborate or delete.

- l. 378-381 The nonlinear relationship of NOx with NO2 TVCD is important, but so are the effects of properly accounting for the background. The fact that spaceborne instruments have a low sensitivity close to the surface (i.e. the averaging kernels) is also important and deserves to be mentioned in this discussion.

Technical comments:

- in the title, "tend" should be "trend"

- abstract line 15, add the word "bottom-up" (or "estimated') before "anthropogenic"

- l. 89 "mechanistic" (not "mechanical")

- l. 107 replace "measurements" by "sensors"

- l. 109 add "instrument" after "SCIAMACHY"

- l. 116 "These instruments measure transmitted, backscattered, and reflected radiation" is unclear

- l. 126 "OMINO2" (not OMNO2")

- l. 134 "choose" not "chose" (I guess)

- l. 208 add "the" before "model simulation"

- l. 279 -280 "sensitivities (...) to different anthropogenic NOx emissions over the CONUS" is confusing, please rephrase

- l. 325 insert "the" before "decreasing rates"

- References : use journal abbreviations, e.g. Atmos. Environ., etc.

- caption of Figure 5, line 672: specify the year (and month?) of the anthropogenic emissions used to define the groups
* * *

---

## Author Comment (AC3) · 10 Oct 2019

The comment was uploaded in the form of a supplement:
https://www.atmos-chem-phys-discuss.net/acp-2019-472/acp-2019-472-AC3-
supplement.pdf

---

## Author Comment (AC4) · 10 Oct 2019

**Response to reviewer #1**

Thank you for the careful and thorough reading of this manuscript and your thoughtful comments and suggestions. Our responses follow the reviewer's comments (in *Italics*).

*General comments:*

*There are a number of recent papers on the topic of NOx emission trends in the United States as observed from space and as compared to predictions from models. The papers raise issues about the emission models, about the resolution of measurements and models needed to derive accurate trends, about interpretation of satellite observations including whether and how the regional background is included in the trend analysis, and whether the lifetime of NOx is also changing with time affecting interpretation of temporal trends. The analysis in this paper focuses on nonlinearities in chemistry which is related to the question of chemical lifetime. The analysis in the paper seems solid and the discussion and conclusions try to put the paper in context of the recent literature.*

*I recommend the abstract be revisited in light of the discussion and conclusions as nowwritten.*

**Reply:**

Thank you for your suggestion. We listed the factors affecting the nonlinear relationships among anthropogenic $NO_x$ emissions, $NO_2$ surface concentrations, and $NO_2$ TVCDs in Lines 17 - 18 in the revised manuscript. Not only chemistry and background sources but also physical processes, such as transport, contribute to the nonlinearities.

*I also recommend the authors consider whether they can make some more general conclusions about the role of nonlinearities that are the focus of their work as a guide to future research. For example does this research help push forward the conversation about the model resolution needed to describe NOx to a specified accuracy? Other papers suggest that 36km might not be sufficient for the absolute accuracy the authors are trying to achieve. On the other hand there might be cancellation of errors in computation of trends that allows use of lower resolution for questions about trends?*

**Reply:**

We think 36 km is sufficient for the regional analysis in this study. A higher resolution model result will not change the nonlinearity discussion in section 3.1. The low-anthropogenic-$NO_x$ emission regions are more sensitive to various factors, such as lighting and soil $NO_x$ emissions and transport, than high-anthropogenic-$NO_x$ emission regions. The critical thing in trend analysis using satellite data directly is therefore to use the data over high-anthropogenic-$NO_x$ emission regions and avoid low-anthropogenic-$NO_x$ emission regions. Therefore, the favorable resolution depends on the emission distributions of the study area. The model analysis used here is only to show the problems associated with using data over the low-anthropogenic-$NO_x$ emission regions. Silvern et al. (2019) used modeling results with a resolution of $0.5° \times 0.625°$ and shown that the tropospheric $NO_x$ lifetime decreased from 8.1 hours to 7.7 hours from 2005 – 2017. When using high-resolution simulations, suggested by Valin et al. (2011), the required accuracy on the anthropogenic emission distribution is much higher than 36 km. Our model results using 4- and 36-km resolutions indicate that the errors of 4-km

NO$_x$ emission distribution are significant and need to be accounted for in modeling

analysis. It is beyond what is of interest in this study.

**References**

Silvern, R. F., Jacob, D. J., Mickley, L. J., Sulprizio, M. P., Travis, K. R., Marais, E. A., Cohen, R. C., Laughner, J. L., Choi, S., Joiner, J., and Lamsal, L. N.: Using satellite observations of tropospheric NO$_2$ columns to infer long-term trends in US NO$_x$ emissions: the importance of accounting for the free tropospheric NO$_2$ background, Atmos. Chem. Phys., 19, 8863-8878, 10.5194/acp-19-8863-2019, 2019.

Valin, L. C., Russell, A. R., Hudman, R. C., and Cohen, R. C.: Effects of model resolution on the interpretation of satellite NO$_2$ observations, Atmos. Chem. Phys., 11, 11647-11655, 10.5194/acp-11-11647-2011, 2011.

---

## Author Comment (AC5) · 10 Oct 2019

The comment was uploaded in the form of a supplement:
https://www.atmos-chem-phys-discuss.net/acp-2019-472/acp-2019-472-AC5-supplement.pdf

2019.

---

## Author Comment (AC6) · 10 Oct 2019

**Response to reviewer #2**

We thank the reviewer for careful and thorough reading of this manuscript and the thoughtful comments and suggestions. Our answers follow the reviewer's comments (in *Italics*).

*General comments:*

*There has been many studies in the last years on the recent trends of NOx emissions over the U.S., the main motivation being the apparent important change in NO2 column trend since 2010, which obviously requires careful analysis using the available data as well as using models. The present study is useful, as it clearly shows that there is no significant discrepancy between the NEI emission trends and the different NO2 (surface and column) data, when considering only urban areas. The paper discusses the non-linear relationship between NOx emissions and NO2 abundances. Model calculations using REAM at 36kmx36 km are used to illustrate this point and show that the feedbacks are much stronger at low-NOx than at high-NOx. Although the relevance of NOx natural emissions (which obviously do not have the same trends as the anthropogenic component) is mentioned, the paper does not dwell on it.*

*In fact, and this is my main comment, I think clarifications are needed in order to sort out the respective roles of chemical non-linearities and the existence of the background. Both natural emissions and chemical non-linearities play their largest role during summer over rural areas, and more so in the free troposphere than near the surface. But it is not entirely clear from the paper how much these two main factors contributed to*

*the apparent discrepancy between the different sets of trends. This should be clarified.*

**Reply:**

Thank you for your suggestions. Since in our trend analyses in Section 3.3, we chose urban regions with small $\beta$ and $\gamma$ values and had minimized the impacts of chemical nonlinearity and background sources on inferring anthropogenic $NO_x$ emissions from satellite datasets. Silvern et al. (2019) also show that lightning $NO_x$ and the lifetime of tropospheric $NO_x$ have no significant trend signals from 2005 – 2017. We think you ask which factors affect $\beta$ and $\gamma$ more.

Due to the interactions among $NO_x$ emissions, chemistry, and physical processes, it is difficult to completely and accurately separate the effects of all factors to $\beta$ and $\gamma$ values. Here, we estimated the impact of background sources and non-emission factors (transport, chemistry, and wet and dry depositions) on $\beta$ and $\gamma$ values and added two supplement figures (Figures S6 and S7) in Lines 105 – 143 in the revised supplement figure file. The supplement figure citation was updated in the manuscript. We also added "transport" in Line 241. Figures S6 and S7 show that the contributions of both background sources and non-emission factors to $\beta$ and $\gamma$ values are much more significant in low-anthropogenic-$NO_x$ emission regions than high-anthropogenic-$NO_x$ emission regions. In general, non-emission factors contribute more to the nonlinearity than background sources in low-anthropogenic-$NO_x$ emission regions (Figures S7c and S7d) except for the first bin (of low local emissions) where background sources contribute more to the nonlinearity than non-emission factors at 10:00 – 11:00 LT. We added the discussion about the contributions of the two factors to $\beta$ and $\gamma$ values in Lines

231 – 237 and Lines 257 – 264.

*Also, although the paper mentions the use of observed $NO_3^-$ deposition trends to further support the declining trend of NOx emissions, it would be useful to incorporate more explicitly this information in the discussion.*

**Reply:**

We mentioned nitrate wet deposition fluxes in the introduction in Lines 43 - 47 in the revised manuscript to support the decrease of $NO_x$ emissions from the mid-2000s to the 2010s based on previous researches. Now we added a new supplement Figure S1 based on the National Acid Deposition Program (NADP) observations over the CONUS in Lines 75 – 79 in the revised supplement figure file, which shows a decrease (~ 30% - ~ 40%) of nitrate wet deposition fluxes from 2003 – 2017. In addition, we mentioned in Lines 376 - 378 in the revised manuscript that Silvern et al. (2019) used nitrate wet deposition fluxes in their analyses. Unlike the study of Silvern et al., which have multiyear simulation results and can compare model results with nitrate wet deposition flux observations, we ran 1-month simulation to show the nonlinearities among anthropogenic $NO_x$ emissions, $NO_2$ surface concentrations, and $NO_2$ TVCDs to support the separation between urban and rural regions in our trend analyses. As discussed in Silvern et al. (2019), nitrate wet deposition fluxes are affected by both boundary $NO_x$ and free-tropospheric $NO_x$, and most nitrate wet deposition flux sites are in rural regions. We didn't find any significant improvement from rural to urban regions when comparing nitrate wet deposition fluxes with coincident OMI-QA4ECV $NO_2$ TVCDs as shown in Figure R1 (Urban: TVCD = $1.13 \times$ NADP + 0.13, $R^2$ = 0.84; Rural: TVCD =

$1.49 \times$ NADP – 0.11, $R^2 = 0.82$), which is a key point of our study. We suggest reading Silvern et al. paper for more details about nitrate wet deposition fluxes.

[Figure]

Figure R1. Relative annual variations of NADP nitrate wet deposition fluxes and coincident OMI-QA4ECV $NO_2$ TVCD in each season from 2005 – 2017 for urban (left panel) and rural (right panel) regions. The observation data are scaled by the corresponding 2005 values. Black and red lines denote NADP nitrate wet deposition fluxes and OMI-QA4ECV $NO_2$ TVCDs, respectively. Shading in a lighter color is added

to show the standard deviation of the results.

***Additional (minor) comments:***

*- l. 34, the total of 0.24 Tg N for natural NOx emissions seems to be very low, where does it come from? I don't think NEI2014 provides this information. Please provide separately the soil, biomass burning and lightning emission information.*

**Reply:**

Thank you for your suggestion. Unlike the natural $NO_x$ sources from Seinfeld and Pandis (2016), which includes both lightning and soil $NO_x$ emissions, NEI2014 only provides soil $NO_x$ emissions calculated by the Biogenic Emission Inventory System (BEIS) but no lightning $NO_x$ emissions (EPA, 2018). The 0.24 Tg N of natural $NO_x$ emissions refers to soil $NO_x$ emissions. We changed "anthropogenic and natural $NO_x$" to "anthropogenic and soil $NO_x$" in Line 35. And we provided soil and lightning $NO_x$ emissions from REAM over the CONUS in July 2011 in Lines $112 - 115$ in the revised manuscript.

*- l. 64-65: there are earlier references for the effect of non-linearities on NO2 trends*

**Reply:**

Yes, we added a citation of Lamsal et al. (2011). Please see Lines $65 - 66$ in the revised manuscript.

*- section 2.1 on REAM. What is the model domain?*

**Reply:**

The model domain is shown in Figure 3, covering the CONUS. We added "the model domain of which is shown in Figure 3" in Line 95 in the revised manuscript to show the model domain.

*- l. 96: How is meteorology constrained by NCEP?*

**Reply:**

NCEP CFSv2 datasets provide initial and boundary conditions for our WRF simulation.

*- l. 100-102 it's a detail, but it seems a little strange that weekday emissions are based on NEI while weekend values are reduced. Isn't NEI an average?*

**Reply:**

Our NEI2011 emission inventory is from PNNL and has an initial horizontal resolution of 4 km. We re-gridded it to 36 km. The emission inventory was calculated by using the Sparse Matrix Operator Kernel Emissions (SMOKE) model which could produce hourly emissions for each day, thus could separate weekdays and weekends. We obtained only averaged weekday emissions from PNNL but no weekend emissions. Therefore, we scaled the weekend emissions based on previous studies (Beirle et al., 2003; Boersma et al., 2009; Choi et al., 2012; DenBleyker et al., 2012; Herman et al., 2009; Judd et al., 2018; Kaynak et al., 2009; Kim et al., 2016) and our model evaluations with observations. Currently, GEOS-Chem and CMAQ provide hourly anthropogenic emissions for each day for NEI2011 and NEI2014, respectively, such as NEI2014v2 at

https://www.acom.ucar.edu/Models/EPA/cmaq_cb6/all/. NEI2005 at

ftp://aftp.fsl.noaa.gov/divisions/taq/emissions_data_2005/ also provides weekday,

Saturday, and Sunday emissions separately.

*- l. 105 what about lightning emissions?*

**Reply:**

We described the method to calculate lightning $NO_x$ emissions in Lines $107 - 112$ in the

revised manuscript.

*- l. 148-149 the requirement that RCI > 50% is quite strict. What happens to the trends*

*when you change that?*

**Reply:**

When we changed the criterion to $RCI < 100\%$, about 17% of seasonal data were

removed. The following Figure R2 is for $RCI < 100\%$. In Figure R3, we included all

seasonal data with any RCI values. Generally, the trends of satellite $NO_2$ TVCDs over

urban regions are still consistent with the trends of EPA $NO_x$ emissions and surface $NO_2$

measurements in both Figure R2 and Figure R3, although there are some differences

among Figure R2, Figure R3, and Figure 6 in the main manuscript. It emphasizes the

selection of urban regions in trend analyses. Here, we would like to keep the $RCI < 50\%$

criterion in the main manuscript as it removes the effects of outliers.

[Figure]

Figure R2. Same as Figure 6 in the main manuscript, but for RCI < 100%.

[Figure]

Figure R3. Same as Figure 6 in the main manuscript, but for all seasonal data with any RCI values.

*- l. 184 how many measurements are rejected from this conditions on RCI?*

**Reply:**

For surface concentrations, due to the completeness and stability of surface measurements, almost all seasonal averages (98.5%) satisfy the RCI < 50% criterion.

We added the information in Lines 194 – 195 in the revised manuscript.

*- l. 202 Are the β and γ calculated based on total emissions with or without lightning emissions? Lightning contributes significantly to the total column, but very little to surface concentrations (in part due to the vertical dependence of spaceborne instruments sensitivity).*

**Reply:**

Yes. Surface $NO_2$ concentrations are not much affected by $NO_x$ in the free troposphere, which $NO_2$ in the free troposphere is an important component of $NO_2$ TVCD. We have discussed it in Lines 248 – 251. Both β and γ are calculated based on the emissions without lightning. The lifetime of lightning $NO_x$ in the free troposphere is much longer than that in the boundary layer. As mentioned above, we added two supplement figures (Figures S6 and S7) to evaluate the contributions of different factors to β and γ values.

*- l. 229 "such as NOx transport from nearby regions" this is surprising since the calculated sensitivities were said to be purely local*

**Reply:**

In Lines 225 – 226 in the revised manuscript, we said, "Using this procedure, the effects of anthropogenic $NO_x$ emission reduction were localized". It doesn't mean that transport effect is eliminated. Let's think about a simple example, to calculate β and γ values for a single grid cell "A", we only need to adjust the $NO_x$ emissions of "A" but keep all other grid cells the same as before. By comparing two simulations, one with the original

emissions, the other one with grid cell "A" adjusted, we can obtain the β and γ values of "A". Here, only the $NO_x$ emissions of "A" are reduced in the adjusted simulation, and other grid cells are unchanged, so the emission reduction effect is localized. But transport still makes effects. Outfluxes from "A" to nearby grid cells will be different from the original simulation, as $NO_x$ concentrations in "A" change. Our method described in Lines 216 – 225 in the revised manuscript can simulate the above procedure simultaneously for all grid cells and save computing time. This idea is different from a method widely used in previous studies by comparing one simulation with original emissions and the other one with emission reductions for all grid cells, where not only outfluxes from "A" change but also influxes to "A" are different from the original simulation. That is to say, the emission reductions of nearby grids are affecting grid cell "A", which cannot be used to calculate local β and γ values.

*l. 234 there is no "transport effect". β and γ are closer to 1 at 10-11 LT (compred to 13-14 LT) because of the weaker chemical losses.*

**Reply:**

As we explained in the above answer, there are transport effects in the calculation of β and γ. In Line 234 in the original manuscript (Lines 251 – 253 in the revised manuscript), we were talking about the uncertainties of β and γ in each bin, and generally we don't have enough evidence from Figure 2 to show that β and γ are closer to 1 at 10-11 LT compared to 13-14 LT.

*l. 242 I suppose the "urban" definition depends on anthropogenic NOx emissions on a*

*specific year (and month maybe). This should be specified.*

**Reply:**

Thank you for your suggestion. Yes, the definition is based on NEI2011, as described in Section 2.1, which provides annual average emissions for 2011 weekdays. We changed "anthropogenic $NO_x$ emissions " to "anthropogenic $NO_x$ emissions from NEI2011" to make it clear. Please see Lines 268 – 269 in the revised manuscript.

*l. 330-332 Note that only 22 AQS sites (out of 179) are rural. Therefore, is the difference between this study and the results of Lamsal et al. and Jiang et al. really due to the selection of urban sites?*

**Reply:**

Figure R4 shows the comparison between mean $NO_2$ concentrations from AQS urban sites and those from all (urban + rural) AQS sites, and there is no significant difference. Silvern et al. (2019) suggested that Jiang et al. (2018) included those sites with incomplete measurement records, which might be the reason why Jiang et al. (2018) had lower slowdown magnitude compared to our study (Table 1 in the main manuscript) and Silvern et al. (2019). The decreasing rates of AQS $NO_2$ concentrations in Lamsal et al. (2015) (Table 1 in the main manuscript) are smaller than our study and Silvern et al. (2019) (2005 – 2009: $-6.6 \pm 1.2\%$ $a^{-1}$; 2011 – 2015: $- 4.5 \pm 1.7\%$ $a^{-1}$), which might also be partly due to their different data processing procedure. We changed the original sentence, please see Lines 356 – 361 in the revised manuscript.

[Figure]

Figure R4. Relative annual variations of mean $NO_2$ surface concentrations from AQS sites. Black lines denote mean concentrations for only AQS urban sites, while red lines are for all AQS sites, including both rural and urban. The mean $NO_2$ concentrations are scaled by the corresponding 2003 values. The left column is for $NO_2$ concentrations at 10:00 – 11:00 LT, and the right column is for 13:00 – 14:00 LT.

*l. 349-350 the sentence "They also identified model biases (...) natural emissions" is unclear, please either elaborate or delete.*

**Reply:**

Silvern et al. (2019) shown that GEOS-Chem v11-02c underestimated $NO_2$ concentrations in the free troposphere compared to aircraft observations and satellite cloud-slicing results, which they thought was the reason why GEOS-Chem simulation results couldn't capture satellite $NO_2$ TVCD trends. We changed "natural emissions" to "missing natural emissions in the free troposphere" in Line 379 in the main manuscript.

*- l. 378-381 The nonlinear relationship of NOx with NO2 TVCD is important, but so are the effects of properly accounting for the background. The fact that spaceborne instruments have a low sensitivity close to the surface (i.e. the averaging kernels) is also important and deserves to be mentioned in this discussion.*

**Reply:**

Thank you for the suggestion. In this study, when we talk about nonlinearity ($\beta$ and $\gamma$), we always mean any chemical and physical processes affecting the $NO_2$ TVCD and $NO_2$ surface concentrations, such as soil $NO_x$ in the boundary layer and lightning $NO_x$ in the free troposphere, chemistry, transport effect, and wet-dry depositions. We added other nonlinear factors in Lines 204 – 205 in the revised manuscript to make it clear. In Section 3.1, as mentioned above, now we have more discussion about the contributions of different factors to $\beta$ and $\gamma$ values. The low sensitivity of satellite sensors to the surface $NO_x$ indeed emphasizes the selection of urban regions in inferring anthropogenic $NO_x$ emissions from satellite datasets with more $NO_x$ in the lower atmosphere compared to free troposphere to make the satellite signal meaningful to anthropogenic $NO_x$

emissions, but it is more related to the satellite measurement uncertainties which we have talked about in Lines 152 – 154 in the revised manuscript. We recommend reading Silvern et al. (2019) for more details about the vertical sensitivity of satellite sensors to $NO_2$ distributions.

*Technical comments:*

*- in the title, "tend" should be "trend"*

**Reply:**

Thanks. We corrected it.

*- abstract line 15, add the word "bottom-up" (or "estimated') before "anthropogenic"*

**Reply:**

The results shown in Lines 14 – 19 are based on the 1-month REAM simulation, where we indeed used the bottom-up NEI2011 emission inventory. However, the conclusions are widely applicable and not limited to NEI2011 or any other bottom-up emission inventories.

*- l. 89 "mechanistic" (not "mechanical")*

**Reply:**

Thanks. We corrected it. Please see Line 90 in the revised manuscript.

*- l. 107 replace "measurements" by "sensors"*

**Reply:**

We corrected it. Please see Line 117 in the revised manuscript.

*- l. 109 add "instrument" after "SCIAMACHY"*

**Reply:**

We added it. Please see Line 120 in the revised manuscript.

*- l. 116 "These instruments measure transmitted, backscattered, and reflected radiation"*
*is unclear*

**Reply:**

We changed it to a simple sentence "These instruments measure backscattered solar radiation). Please see Line 127 in the revised manuscript.

*- l. 126 "OMINO2" (not OMNO2")*

**Reply:**

NASA OMI NO$_2$ TVCD products are named as OMNO2. Please refer to https://disc.gsfc.nasa.gov/datasets/OMNO2_V003/summary.

*- l. 134 "choose" not "chose" (I guess)*

**Reply:**

Thanks. We think it would be better to change "re-grid" to "re-gridded" in Line 145 in

the revised manuscript.

*- l. 208 add "the" before "model simulation"*

**Reply:**

Thanks. We added it. Please see Line 220 in the revised manuscript.

*- l. 279 -280 "sensitivities (…) to different anthropogenic NOx emissions over the CONUS" is confusing, please rephrase*

**Reply:**

Yes. We changed it to "We further investigate OMI-QA4ECV $NO_2$ TVCD relative annual variations from 2005 - 2017 over the regions with different anthropogenic $NO_x$ emissions in Figure 5." Please see Lines 305 – 307 in the revised manuscript.

*- l. 325 insert "the" before "decreasing rates"*

**Reply:**

Thanks. We added it. Please see Line 353 in the revised manuscript.

*- References : use journal abbreviations, e.g. Atmos. Environ., etc.*

**Reply:**

Yes, we corrected it.

*- caption of Figure 5, line 672: specify the year (and month?) of the anthropogenic*

*emissions used to define the groups*

**Reply:**

Yes, we added it. Please see Line 705 in the revised manuscript.

**References**

Beirle, S., Platt, U., Wenig, M., and Wagner, T.: Weekly cycle of $NO_2$ by GOME measurements: A signature of anthropogenic sources, Atmos. Chem. Phys., 3, 2225-2232, 10.5194/acp-3-2225-2003, 2003.

Boersma, K. F., Jacob, D. J., Trainic, M., Rudich, Y., De Smedt, I., Dirksen, R., and Eskes, H. J.: Validation of urban $NO_2$ concentrations and their diurnal and seasonal variations observed from the SCIAMACHY and OMI sensors using in situ surface measurements in Israeli cities, Atmos. Chem. Phys., 9, 3867-3879, 10.5194/acp-9-3867-2009, 2009.

Choi, Y., Kim, H., Tong, D., and Lee, P.: Summertime weekly cycles of observed and modeled $NO_x$ and $O_3$ concentrations as a function of satellite-derived ozone production sensitivity and land use types over the Continental United States, Atmos. Chem. Phys., 12, 6291-6307, 10.5194/acp-12-6291-2012, 2012.

DenBleyker, A., Morris, R. E., Lindhjem, C. E., Parker, L. K., Shah, T., Koo, B., Loomis, C., and Dilly, J.: Temporal and Spatial Detail in Mobile Source Emission Inventories for Regional Air Quality Modeling, 2012 International Emission Inventory Conference, Florida, U.S., August 13 - 16, 2012, 2012.

EPA: 2014 National Emissions Inventory, version 2 - Technical Support Document, Research Triangle Park, North Carolina, 414, 2018.

Herman, J., Cede, A., Spinei, E., Mount, G., Tzortziou, M., and Abuhassan, N.: $NO_2$ column amounts from ground-based Pandora and MFDOAS spectrometers using the direct-Sun DOAS technique: Intercomparisons and application to OMI validation, J. Geophys. Res.-Atmos., 114, 10.1029/2009JD011848, 2009.

Jiang, Z., McDonald, B. C., Worden, H., Worden, J. R., Miyazaki, K., Qu, Z., Henze, D. K., Jones, D. B. A., Arellano, A. F., and Fischer, E. V.: Unexpected slowdown of US pollutant emission reduction in the past decade, Proc. Natl. Acad. Sci. U.S.A., 201801191, 10.1073/pnas.1801191115, 2018.

Judd, L. M., Al-Saadi, J. A., Valin, L. C., Pierce, R. B., Yang, K., Janz, S. J., Kowalewski, M. G., Szykman, J. J., Tiefengraber, M., and Mueller, M.: The Dawn of Geostationary Air Quality Monitoring: Case Studies from Seoul and Los Angeles, Front. Environ. Sci., 6, 85, 10.3389/fenvs.2018.00085, 2018.

Kaynak, B., Hu, Y., Martin, R. V., Sioris, C. E., and Russell, A. G.: Comparison of weekly cycle of $NO_2$ satellite retrievals and $NO_x$ emission inventories for the continental United States, J. Geophys. Res.-Atmos., 114, 10.1029/2008JD010714, 2009.

Kim, S. W., McDonald, B., Baidar, S., Brown, S., Dube, B., Ferrare, R., Frost, G., Harley, R., Holloway, J., and Lee, H. J.: Modeling the weekly cycle of $NO_x$ and CO emissions and their impacts on $O_3$ in the Los Angeles-South Coast Air Basin during the CalNex 2010 field campaign, J. Geophys. Res.-Atmos., 121, 1340-1360, 10.1002/2015JD024292, 2016.

Lamsal, L. N., Martin, R. V., Padmanabhan, A., Van Donkelaar, A., Zhang, Q., Sioris, C. E., Chance, K., Kurosu, T. P., and Newchurch, M. J.: Application of satellite observations for timely updates to global anthropogenic $NO_x$ emission inventories, Geophys. Res. Lett., 38, 10.1029/2010GL046476, 2011.

Lamsal, L. N., Duncan, B. N., Yoshida, Y., Krotkov, N. A., Pickering, K. E., Streets, D. G., and Lu, Z.: US $NO_2$ trends (2005–2013): EPA Air Quality System (AQS) data versus improved observations from the Ozone Monitoring Instrument (OMI), Atmos. Environ., 110, 130-143, 10.1016/j.atmosenv.2015.03.055, 2015.

Seinfeld, J. H., and Pandis, S. N.: Atmospheric chemistry and physics: from air pollution to climate change, John Wiley & Sons, Inc, Hoboken, New Jersey, 2016.

Silvern, R. F., Jacob, D. J., Mickley, L. J., Sulprizio, M. P., Travis, K. R., Marais, E. A., Cohen, R. C., Laughner, J. L., Choi, S., Joiner, J., and Lamsal, L. N.: Using satellite observations of tropospheric $NO_2$ columns to infer long-term trends in US $NO_x$ emissions: the importance of accounting for the free tropospheric $NO_2$ background, Atmos. Chem. Phys., 19, 8863-8878, 10.5194/acp-19-8863-2019, 2019.

---

## Author Comment (AC7) · 10 Oct 2019

[revised manuscript text omitted]

**Figure Captions**

Figure S1. Annual variation of $NO_3^-$ wet deposition fluxes for each season from 2003 – 2017. The fluxes were scaled by the corresponding values in 2003. Shaded regions denote standard deviations. Monthly $NO_3^-$ wet deposition observations are obtained from https://nadp.slh.wisc.edu/data/NTN/ntnAllsites.aspx (last access, September 29, 2019).

Figure S2. Comparison between original EPA anthropogenic $NO_x$ emissions and updated EPA anthropogenic $NO_x$ emissions with the newest Continuous Emission Monitoring Systems (CEMS) measurements.

Figure S3. Daily OMI $NO_2$ TVCDs for July 2011 (a) and 2012 (b) in Atlanta (33.755° N, 84.39° W). Black circles are weekday values, and red circles are weekend values. We find significant daily variations of $NO_2$ TVCD from (a) and (b). The number of available measurements in July 2011 is much less than July 2012. We find clear larger $NO_2$ TVCD values on weekdays than on weekends in July 2011, but the difference between weekday and weekday TVCDs in July 2012 are not so obvious.

Figure S4. Hourly averaged ratios of FEM (a) and CAPS (b) to FRM $NO_2$ measurements in each season, respectively. The FEM/FRM ratios are computed from coincident FRM and FEM measurements from 2013 – 2015 at 4 sites. The CAPS/FRM ratios are calculated based on coincident CAPS and FRM data from 2015 – 2016 at 3 sites.

Figure S5. Annual variations of AQS $NO_2$ surface concentrations at different hours on weekdays in spring (a, b), summer (c, d), autumn (e, f), and winter (g, h). Left panels show absolute $NO_2$ concentrations, and right panels are their relative variations normalized to 2011. To conduct reliable and consistent comparisons, we only used monitoring sites satisfying the seasonal *RCI* < 50% and continuity criteria on weekdays from 2003 – 2017.

Figure S6. Distributions of (a) $NO_2$ TVCD fraction that is in the boundary layer ($< 2810$ m) at 13:00 – 14:00, (b) $NO_2$ TVCD fraction in the boundary layer ($< 1290$ m) at 10:00 – 11:00, (c) the fraction of soil $NO_x$ emissions in all surface sources (anthropogenic + soil) on weekdays for July 2011. As the lifetime of $NO_2$ in the free troposphere (several days ~ 2 weeks) is much longer than that in the boundary layer (~ 10 hours), local lightning $NO_x$ emissions cannot represent $NO_2$ VCDs in the free troposphere. In this study, we apply $NO_2$ VCD in the free troposphere to analyze the impact of lighting $NO_x$ on the nonlinear relationships between anthropogenic $NO_x$ emissions and $NO_2$ TVCDs and use lightning $NO_x$ and $NO_2$ VCD in the free troposphere interchangeably in the following.

Figure S7. (a) Distributions of the fractions of surface $NO_x$ emissions emitted by soil ("SoilNO$_x$"), the portions of $NO_2$ TVCDs in the boundary layer ("PBLVCD"), and the fractions of $NO_2$ TVCDs from anthropogenic $NO_x$ emissions ("AnthroVCD") as functions of NEI2011 anthropogenic $NO_x$ emissions at 13:00 – 14:00 LT on weekdays for July 2011 over the CONUS. The fraction of $NO_2$ TVCDs from anthropogenic $NO_x$ emissions is equal to $\left(1 - \frac{E_{soil}}{E_{soil}+E_{anthropogenic}}\right) \times \left(\frac{TVCD_{boundary}}{TVCD_{boundary}+TVCD_{free}}\right)$, where $E_{soil}$ denotes soil $NO_x$ emissions, $E_{anthropogenic}$ denotes anthropogenic $NO_x$ emissions, $TVCD_{boundary}$ denotes $NO_2$ TVCDs in the boundary layer, and $TVCD_{free}$ denotes $NO_2$ TVCDs in the free troposphere. The calculated data are grouped into 9 bins as in Figure 2. (b) Same as (a), but for 10:00 – 11:00 LT. (c) Distributions of $\beta_{Emis}$, $\gamma_{Emis}$, $\beta$, and $\gamma$ as functions of anthropogenic $NO_x$ emissions at 13:00 – 14:00 LT on weekdays for July 2011 over the CONUS. $\beta$ and $\gamma$ are the same as Figure 2. $\beta_{Emis}$ and $\gamma_{Emis}$ denote $\beta$ and $\gamma$ values when no other factors are taken into consideration except for soil $NO_x$ emissions, anthropogenic $NO_x$ emissions, and $NO_2$ in the free troposphere. $\beta_{Emis} = \frac{15\%}{15\% \times \left(\frac{E_{anthropogenic}}{E_{anthropogenic}+E_{soil}}\right)\left(\frac{TVCD_{boundary}}{TVCD_{boudnary}+TVCD_{free}}\right)} = \left(\frac{E_{anthropogenic}+E_{soil}}{E_{anthropogenic}}\right)\left(\frac{TVCD_{boundary}+TVCD_{free}}{TVCD_{boundary}}\right)$, and $\gamma_{Emis} = \dfrac{15\%}{15\% \times \left(\dfrac{E_{anthropogenic}}{E_{anthropogenic} + E_{soil}}\right)} = \left(\dfrac{E_{anthropogenic} + E_{soil}}{E_{anthropogenic}}\right)$. It is noteworthy that here we assume no interactions between the boundary layer and the free troposphere, boundary $NO_x$ are only related to soil and anthropogenic $NO_x$ emissions, and lightning $NO_x$ only affect $NO_2$ in the free troposphere. The assumptions are reasonable as the time scale (~ 1 week) of the interactions between the boundary layer and the free troposphere are much longer than $NO_x$ lifetime in the boundary layer, and in this study, only a small fraction of lightning $NO_x$ is distributed into the boundary layer. Therefore, $\beta_{Emis}$ and $\gamma_{Emis}$ roughly represent the contributions of background sources (lightning $NO_x$ and soil $NO_x$) to $\beta$ and $\gamma$ values. The differences between $\beta$ ($\gamma$) and $\beta_{Emis}$

($\gamma_{Emis}$) indicate the contribution of non-emission factors to $\beta$ ($\gamma$) values, such as chemistry, transport, and dry and wet depositions. (d) Same as (c), but for 10:00 – 11:00 LT. From this figure, we find that both background sources (lightning $NO_x$ + soil $NO_x$) and non-emission factors are important when considering the nonlinear relationships among $NO_x$ emissions, $NO_2$

surface concentrations, and $NO_2$ TVCDs in low-anthropogenic-$NO_x$ emission regions.

Figure S8. Same as Figure 4, but for AQS $NO_2$ surface concentrations and coincident GOME-

2A $NO_2$ TVCD data during 2008 – 2016.

Figure S9. Relative variations of OMI-QA4ECV $NO_2$ TVCD data for urban regions (black lines)

and the whole CONUS (red lines) from 2005 – 2017 in 4 seasons.

[Figure]

Figure S1. Annual variation of $NO_3^-$ wet deposition fluxes for each season from 2003 – 2017. The fluxes were scaled by the corresponding values in 2003. Shaded regions denote standard deviations. Monthly $NO_3^-$ wet deposition observations are obtained from https://nadp.slh.wisc.edu/data/NTN/ntnAllsites.aspx (last access, September 29, 2019).

[Figure]

Figure S2. Comparison between original EPA anthropogenic $NO_x$ emissions and updated EPA

anthropogenic $NO_x$ emissions with the newest Continuous Emission Monitoring Systems (CEMS) measurements.

[Figure]

Figure S32. Daily OMI NO$_2$ TVCDs for July 2011 (a) and 2012 (b) in Atlanta (33.755° N, 84.39°

W). Black circles are weekday values, and red circles are weekend values. We find significant daily variations of NO$_2$ TVCD from (a) and (b). The number of available measurements in July

2011 is much less than July 2012. We find clear larger NO$_2$ TVCD values on weekdays than on weekends in July 2011, but the difference between weekday and weekday TVCDs in July 2012

are not so obvious.

[Figure]

Figure S43. Hourly averaged ratios of FEM (a) and CAPS (b) to FRM NO$_2$ measurements in each season, respectively. The FEM/FRM ratios are computed from coincident FRM and FEM

measurements from 2013 – 2015 at 4 sites. The CAPS/FRM ratios are calculated based on coincident CAPS and FRM data from 2015 – 2016 at 3 sites.

[Figure]

Figure S54. Annual variations of AQS NO$_2$ surface concentrations at different hours on weekdays in spring (a, b), summer (c, d), autumn (e, f), and winter (g, h). Left panels show absolute NO$_2$ concentrations, and right panels are their relative variations normalized to 2011. To conduct reliable and consistent comparisons, we only used monitoring sites satisfying the seasonal *RCI* < 50% and continuity criteria on weekdays from 2003 – 2017.

[Figure]

Figure S6. Distributions of (a) $NO_2$ TVCD fraction that is in the boundary layer (< 2810 m) at

13:00 – 14:00, (b) $NO_2$ TVCD fraction in the boundary layer (< 1290 m) at 10:00 – 11:00, (c) the fraction of soil $NO_x$ emissions in all surface sources (anthropogenic + soil) on weekdays for July

2011. As the lifetime of $NO_2$ in the free troposphere (several days ~ 2 weeks) is much longer than that in the boundary layer (~ 10 hours), local lightning $NO_x$ emissions cannot represent $NO_2$

VCDs in the free troposphere. In this study, we apply $NO_2$ VCD in the free troposphere to analyze the impact of lighting $NO_x$ on the nonlinear relationships between anthropogenic $NO_x$

emissions and $NO_2$ TVCDs and use lightning $NO_x$ and $NO_2$ VCD in the free troposphere interchangeably in the following.

[Figure]

Figure S7. (a) Distributions of the fractions of surface NO$_x$ emissions emitted by soil ("SoilNO$_x$"), the portions of NO$_2$ TVCDs in the boundary layer ("PBLVCD"), and the fractions of NO$_2$ TVCDs from anthropogenic NO$_x$ emissions ("AnthroVCD") as functions of NEI2011 anthropogenic NO$_x$ emissions at 13:00 – 14:00 LT on weekdays for July 2011 over the CONUS. The fraction of NO$_2$ TVCDs from anthropogenic NO$_x$ emissions is equal to $\left(1 - \frac{E_{soil}}{E_{soil}+E_{anthropogenic}}\right) \times \left(\frac{TVCD_{boundary}}{TVCD_{boundary}+TVCD_{free}}\right)$, where $E_{soil}$ denotes soil NO$_x$ emissions,

*$E_{anthropogenic}$* denotes anthropogenic $NO_x$ emissions, *$TVCD_{boundary}$* denotes $NO_2$ TVCDs in the boundary layer, and *$TVCD_{free}$* denotes $NO_2$ TVCDs in the free troposphere. The calculated data are grouped into 9 bins as in Figure 2. (b) Same as (a), but for 10:00 – 11:00 LT. (c) Distributions of $\beta_{Emis}$, $\gamma_{Emis}$, $\beta$, and $\gamma$ as functions of anthropogenic $NO_x$ emissions at 13:00 – 14:00 LT on weekdays for July 2011 over the CONUS. $\beta$ and $\gamma$ are the same as Figure 2. $\beta_{Emis}$ and $\gamma_{Emis}$ denote

$\beta$ and $\gamma$ values when no other factors are taken into consideration except for soil $NO_x$ emissions, anthropogenic $NO_x$ emissions, and $NO_2$ in the free troposphere. $\beta_{Emis} =$

$$\frac{15\%}{15\% \times \left(\frac{E_{anthropogenic}}{E_{anthropogenic}+E_{soil}}\right)\left(\frac{TVCD_{boundary}}{TVCD_{boudnary}+TVCD_{free}}\right)} = \left(\frac{E_{anthropogenic}+E_{soil}}{E_{anthropogenic}}\right)\left(\frac{TVCD_{boundary}+TVCD_{free}}{TVCD_{boundary}}\right),$$

and $\gamma_{Emis} = \dfrac{15\%}{15\% \times \left(\frac{E_{anthropogenic}}{E_{anthropogenic}+E_{soil}}\right)} = \left(\dfrac{E_{anthropogenic}+E_{soil}}{E_{anthropogenic}}\right)$. It is noteworthy that here we assume no interactions between the boundary layer and the free troposphere, boundary $NO_x$ are only related to soil and anthropogenic $NO_x$ emissions, and lightning $NO_x$ only affect $NO_2$ in the free troposphere. The assumptions are reasonable as the time scale (~ 1 week) of the interactions between the boundary layer and the free troposphere are much longer than $NO_x$ lifetime in the boundary layer, and in this study, only a small fraction of lightning $NO_x$ is distributed into the boundary layer. Therefore, $\beta_{Emis}$ and $\gamma_{Emis}$ roughly represent the contributions of background sources (lightning $NO_x$ and soil $NO_x$) to $\beta$ and $\gamma$ values. The differences between $\beta$ ($\gamma$) and $\beta_{Emis}$

($\gamma_{Emis}$) indicate the contribution of non-emission factors to $\beta$ ($\gamma$) values, such as chemistry, transport, and dry and wet depositions. (d) Same as (c), but for 10:00 – 11:00 LT. From this figure, we find that both background sources (lightning $NO_x$ + soil $NO_x$) and non-emission factors are important when considering the nonlinear relationships among $NO_x$ emissions, $NO_2$

surface concentrations, and $NO_2$ TVCDs in low-anthropogenic-$NO_x$ emission regions.

[Figure]

Figure S8. Same as Figure 4, but for AQS NO₂ surface concentrations and coincident GOME-

2A NO₂ TVCD data during 2008 – 2016.

[Figure]

Figure S96. Relative variations of OMI-QA4ECV NO₂ TVCD data for urban regions (black lines)
and the whole CONUS (red lines) from 2005 – 2017 in 4 seasons.

**Inferring the anthropogenic NO$_x$ emission trend over the United States during 2003 - 2017 from satellite observations: Was there a flattening of the emission trend after the Great Recession?**

Jianfeng Li[1], Yuhang Wang[1*]

[1]School of Earth and Atmospheric Sciences, Georgia Institute of Technology, Atlanta, Georgia, USA

[*] *Correspondence to* Yuhang Wang (yuhang.wang@eas.gatech.edu)

**Table Captions**

Table S1. Summary of major satellite instruments for remote sensing of atmospheric $NO_2$ VCD in the past decade

Table S2. Summary of satellite $NO_2$ TVCD products and their retrieval information

Table S3. Selection criteria for satellite $NO_2$ TVCD pixel data

Table S4. Summary of annual trends of AQS $NO_2$ surface concentrations and satellite $NO_2$ TVCD products in each region during different periods

**Table S1. Summary of major satellite instruments for remote sensing of atmospheric NO₂ VCD in the past decade**

| Instrument | Satellite | Launch date | End date | Operator | Equator crossing time (local time) | UV/Vis Spectral range (nm) | Spectral resolution (nm) | Swath length (km) | Nadir pixel resolution (km × km) | Global coverage (days) |
|---|---|---|---|---|---|---|---|---|---|---|
| SCIAMACHY | ENVISAT[1] | 03/01/2002[2] | 04/08/2012[2] | ESA[3] | 10:00[1] | $240 - 805$[4] | $0.24 - 0.48$[4] | 960[5] | $60 \times 30$[5] | 6[5] |
| GOME-2A | MetOp-A[6] | 10/19/2006[6] | in operation | EUMETSAT[7] | 9:30[8] | $240 - 790$[8] | $0.26 - 0.51$[8] | 1920 before Jul. 15th, 2013; 960 after Jul. 15th, 2013[8] | $80 \times 40$ before Jul. 15th, 2013; $40 \times 40$ after Jul. 15th, 2013[8] | 1.5[9] |
| GOME-2B | MetOp-B[6] | 09/17/2012[6] | In operation | EUMETSAT | 9:30[8] | $240 - 790$[8] | $0.26 - 0.51$[8] | 1920[8] | $80 \times 40$[8] | 1.5[9] |
| OMI | EOS-Aura[10] | 07/152004[10] | In operation | NASA | 13:45[10] | $270 - 500$[11] | $0.45 - 1.0$[11] | 2600[11] | $24 \times 13$[11] | 1[11] |

[1] Refer to https://earth.esa.int/web/guest/missions/esa-operational-eo-missions/envisat
[2] Refer to https://en.wikipedia.org/wiki/Envisat
[3] The European Space Agency
[4] Refer to http://www.iup.uni-bremen.de/sciamachy/instrument/performance/index.html
[5] Refer to Boersma et al. (2008), Boersma et al. (2009), and (Lee et al., 2009)
[6] Refer to https://www.eumetsat.int/website/home/Satellites/CurrentSatellites/Metop/index.html
[7] The European Organization for the Exploitation of Meteorological Satellites
[8] Refer to EUMETSAT (2015)
[9] Refer to Lee et al. (2009) and Wang et al. (2017)
[10] Refer to https://aura.gsfc.nasa.gov/
[11] Refer to https://aura.gsfc.nasa.gov/omi.html

**Table S2. Summary of satellite NO$_2$ TVCD products and their retrieval information**

| NO$_2$ TVCD products | Version | Available period | DOAS fitting method | Stratosphere–troposphere separation | Fitting window (nm) | Albedo / reflectance | A priori profiles | Radiative transfer model | Cloud | Uncertainty |
|---|---|---|---|---|---|---|---|---|---|---|
| GOME-2B | TM4NO2A (2.3) | 12/20/2012 – current | Intensity fit[1] | Assimilation of satellite total slant columns in the TM4 model[2,3] | 405 – 465[1] | Climatology albedo from 3 years of OMI data[4] | TM4 (2° × 3°)[2] | DAK[2] | FRESCO+ (Oxygen A-band around 760 nm)[5] | 1.0 × 10$^{15}$ molecules/cm$^2$ + 25%[2] |
| SCIAMACHY | QA4ECV (v1.1) | 08/02/2002 – 04/08/2012 | Optical Density[1,6] | Assimilation of OMI total slant columns in the TM5 - MP model[6,7] | 425 – 465[6] | Climatology albedo based on SCIAMACHY[8] | TM5-MP (1° × 1°)[6] | DAK | FRESCO+ | 35% - 45% over polluted scenes; > 100% over background regions (Pacific Ocean)[6] |
| GOME-2A | QA4ECV (v1.1) | 02/01/2007 – 12/31/2016 | | | 405 – 465[1,6] | Climatology albedo based on GOME-2A[8] | | | FRESCO+ | |
| OMI-QA4ECV | QA4ECV (v1.1) | 10/012004 – Current | | | 405 – 465[1,6] | Climatology albedo from 5 years of OMI data[6] | | | Improved O$_2$-O$_2$ (477 nm)[9] | |
| OMI-NASA | SPv3 | 01/01/2005 – 07/31/2017 | Stepwise intensity fit with monthly averaged solar irradiance spectrum[1,10] | Based on OMI total slant columns over regions with low estimated TVCD contributions (TVCD contributions less than 0.3 × 10$^{15}$ molecules/cm$^2$)[10] | 402 – 465[1,10] | OMI climatology albedo[10] | GMI (1° × 1.25°)[10] | TMORAD[10] | O$_2$-O$_2$ (477 nm)[10,11] | SPv2.1 TVCD has uncertainties of about 30% under clear-sky conditions to about 60% under cloudy conditions[12], and the relative difference between SPv3 and SPv2.1 is less than ~20%[10]. |
| OMI-BEHR[13] | v3.0B | 01/01/2005 – 07/31/2017 | | | | Based on MCD43D BRDF product (for land) and model parameterization (for ocean) | WRF-Chem (12 km) | | | ~ 45% on average[14] |

[1] Refer to Zara et al. (2018)
[2] Refer to Boersma et al. (2011). "TM4" is the Tracer Model, version 4. "DAK" is the Doubling-Adding KNMI (DAK) radiative transfer model.
[3] Refer to Williams et al. (2009)
[4] Refer to Kleipool et al. (2008)
[5] Refer to Wang et al. (2017) and Wang et al. (2008)
[6] Refer to Boersma et al. (2018)
[7] Refer to Williams et al. (2017)
[8] Refer to Tilstra et al. (2017)
[9] Refer to Veefkind et al. (2016)
[10] Refer to Bucsela et al. (2013), Bucsela et al. (2016), Krotkov et al. (2017), and Marchenko et al. (2015). "TMORAD" is the TMOS radiative transfer model.
[11] Refer to Acarreta et al. (2004)
[12] Refer to Lamsal et al. (2014), Oetjen et al. (2013), and Tong et al. (2015)
[13] Refer to Laughner et al. (2018). OMI-BEHR uses the SCD from OMI-NASA SPv3 but updates inputs for the AMF calculation, such as a prior NO$_2$ vertical profiles and surface reflectance. Besides, OMI-BEHR only provides NO$_2$ TVCD over the contiguous
United States (CONUS). As in this study, we used the OMI-NASA datasets archived in the OMI-BEHR product, so we only obtained OMI-NASA datasets extended to July 31, 2017.
[14] Average uncertainty over the CONUS is calculated based on the file from http://behr.cchem.berkeley.edu/behr/BEHR-us-uncertainty.hdf

**Table S3. Selection criteria for satellite NO$_2$ TVCD pixel data**

| NO$_2$ TVCD products | Period | Solar zenith angle | albedo | Cloud radiance fraction | Snow or ice covered | AMFtrop/AMFgeo | Flag for retrieval success | Retrieval quality flag | Rows in swath |
|---|---|---|---|---|---|---|---|---|---|
| GOME-2B | 01/01/2013 – 12/31/2017 | < 80° | <= 0.3 | <= 50% | No | > 0.2 | Yes | | All |
| SCIAMACHY | 01/01/2003 – 12/31/2011 | < 80° | <= 0.3 | <= 50% | No | > 0.2 | Yes | | All |
| GOME-2A | 01/01/2008 – 12/31/2016 | < 80° | <= 0.3 | <= 50% | No | > 0.2 | Yes | | All |
| OMI-QA4ECV[1] | 01/01/2005 – 12/31/2017 | < 80° | <= 0.3 | <= 50% | No | > 0.2 | Yes | | 6 - 21 |
| OMI–NASA[1] | 01/01/2005 – 12/31/2016 | < 80° | <= 0.3 | <= 50% | | | Yes | Yes | 6 – 21 |
| OMI-BEHR[1] | 01/01/2005 – 12/31/2016 | < 80° | <= 0.3 | <= 50% | | | Yes | Yes | 6 - 21 |

[1] Rows 6-21 are selected to remove the anomalies developed in the OMI sensor (Boersma et al., 2018; Zhang et al., 2018).

**Table S4. Summary of annual trends of AQS NO$_2$ surface concentrations and satellite NO$_2$ TVCD products in each region during different periods[1]**

| | | Northeast | | Midwest | | South | | West | |
|---|---|---|---|---|---|---|---|---|---|
| | | AQS site | CONUS | AQS site | CONUS | AQS site | CONUS | AQS site | CONUS |
| AQS NO$_2$ VMR at 13:00 -14:00 | 2003 – 2011 | -6.8 ± 0.7% | | -6.1 ± 1.2% | | -6.6 ± 0.7% | | -7.6 ± 1.2% | |
| | 2011 – 2017 | -8.0 ± 1.2% | | -6.4 ± 0.8% | | -5.8 ± 0.6% | | -7.2 ± 1.6% | |
| AQS NO$_2$ VMR at 10:00 – 11:00 | 2003 – 2011 | -6.6 ± 0.5% | | -5.8 ± 1.5% | | -6.5 ± 1.3% | | -7.1 ± 1.6% | |
| | 2011 – 2017 | -7.6 ± 1.0% | | -6.8 ± 0.5% | | -5.7 ± 0.1% | | -6.1 ± 1.1% | |
| SCIAMACHY | 2003 – 2011 | -17.1 ± 2.7% | -11.0 ± 3.3% | -12.9 ± 6.8% | -6.5 ± 0.8% | -9.1 ± 1.0% | -6.2 ± 1.5% | -9.1 ± 1.8% | -7.0 ± 1.4% |
| | 2011 – 2017 | | | | | | | | |
| GOME2B | 2003 – 2011 | | | | | | | | |
| | 2013 – 2017 | -11.4 ± 3.7% | -10.8 ± 3.9% | -9.9 ± 13.1% | -4.4 ± 27.2% | -8.9 ± 3.0% | -7.5 ± 3.6% | -11.8 ± 3.0% | -10.6 ± 2.3% |
| OMI-QA4ECV | 2005 – 2011 | -14.2 ± 6.3% | -10.6 ± 3.8% | -9.2 ± 4.2% | -8.4 ± 2.8% | -9.2 ± 2.7% | -8.2 ± 1.5% | -10.5 ± 1.6% | -8.7 ± 0.9% |
| | 2011 – 2017 | -18.0 ± 16.2% | -7.6 ± 4.2% | -7.6 ± 3.3% | -7.0 ± 1.7% | -4.8 ± 1.4% | -4.6 ± 1.0% | -6.4 ± 1.4% | -4.8 ± 1.2% |
| OMI-NASA | 2005 – 2011 | -11.8 ± 1.3% | -11.0 ± 1.8% | -10.9 ± 4.8% | -10.0 ± 4.1% | -10.0 ± 3.5% | -9.5 ± 1.9% | -10.2 ± 1.8% | -8.5 ± 0.9% |
| | 2011 – 2016 | -10.0 ± 4.9% | -8.5 ± 3.8% | -13.2 ± 3.2% | -9.2 ± 2.7% | 0.3 ± 19.2% | -8.0 ± 5.5% | -9.0 ± 5.7% | -6.6 ± 3.9% |
| OMI-BEHR | 2005 – 2011 | -11.8 ± 1.8% | -10.9 ± 1.9% | -12.2 ± 7.3% | -9.8 ± 4.4% | -9.5 ± 3.1% | -8.8 ± 2.0% | -9.9 ± 1.1% | -8.2 ± 0.4% |
| | 2011 – 2016 | -8.2 ± 3.4% | -6.6 ± 1.7% | -27.4 ± 24.3% | -8.1 ± 3.0% | -7.2 ± 2.3% | -5.0 ± 1.3% | -13.2 ± 14.5% | -7.0 ± 4.8% |

[1] Annual trends are the averages of regional seasonal trends (e.g, Figure 7).

---

## Author Response (AR2)

Dr. Andreas Richter

Co-Editor

Atmospheric Chemistry and Physics

Oct. 20, 2019

Dear Dr. Richter,

Subject: Revision and resubmission of manuscript #acp-2019-472

Thanks again for your careful reviewing of our manuscript and your suggestions. We have carefully reviewed the comments and have revised the manuscript accordingly. Our responses are given in a point-by-point manner below. In addition, Jianfeng Li has moved to PNNL, and the affiliation was updated.

We tracked all the changes and updated the reference format following the ACP style. And we hope the revised version is now suitable for publication.

Please address all correspondence concerning this manuscript to Dr. Yuhang Wang (yuhang.wang@eas.gatech.edu).

Thanks again for your time.

Sincerely,

Jianfeng Li

School of Earth and Atmospheric Sciences
Georgia Institute of Technology
Ferst Drive
Atlanta, GA, 30332-0340

**Response to Co-editor**

Thank you for a careful and thorough reading of the manuscript and for your thoughtful comments and suggestions. Our answers follow the Co-editor's comments (in *Italics*).

*Comments / Suggestions:*

• *In my opinion, a weakness of this work is the lack of separation between chemical and other factors resulting in non-linearities. In the manuscript, chemical non-linearities are often mentioned but they probably play only a minor role compared to the main factor, the relative contribution of anthropogenic to total NOx emissions in a given grid cell. If that's possible, it would therefore be good to add some information on which fraction of the observed non-linearity is really due to chemical non-linearities.*

**Reply:**

Thank you for your suggestions. As we mentioned before, it is tough to accurately quantitively separate the contributions of each factor to $\beta$ and $\gamma$ values due to their complex interactions during the 3-D model. Here, we qualitatively estimated the chemical nonlinearity by using the chemical lifetime of $NO_x$. We updated Figure S7 in the revised supplement figure file, comparing the chemical lifetimes of $NO_x$ for the standard REAM simulation ("group 1" in Section 3.1 in the main manuscript) and those for the model results from "group 2" with reduced anthropogenic $NO_x$ emissions. Since the $NO_x$ chemical lifetimes change little, we can state that chemical nonlinearity does not contribute significantly to the nonlinear relationships in low-anthropogenic-$NO_x$ emission regions where background sources and transport strongly affect $\beta$ and $\gamma$ values.

In high-anthropogenic-$NO_x$ emission regions, the impact of background sources and transport effects on $\beta$ and $\gamma$ values is much weaker than that in low-anthropogenic-$NO_x$ emission regions; therefore, lifetime change should be taken into consideration for more careful analyses but not for this study due to the sharp contrast between rural and urban regions.

Although Figure S7 indeed gives the relative changes of $NO_x$ chemical lifetimes, the relative changes of chemical lifetimes are not directly related to $\frac{\Delta\Omega}{\Omega}$ and $\frac{\Delta c}{c}$ in Equations (1) and (2) in the main manuscript. The following gives a simple example.

We assume that $NO_x$ emission $E_0$ is emitted at time 0, and the chemical lifetime of $NO_x$ is $\tau$. The decay of $E_0$ against chemistry is described below.

$$\frac{dE}{dt} = -\frac{1}{\tau}E$$
$$E = E_0 e^{-\frac{1}{\tau}t}$$

(1)

For another chemical lifetime of $NO_x$, assuming $\tau_1 = 1.1 \times \tau$, we have

$$E_1 = E_0 e^{-\frac{1}{\tau_1}t}$$
$$\frac{E}{E_1} = e^{t\left(\frac{1}{\tau_1}-\frac{1}{\tau}\right)} = e^{t\left(\frac{\tau-\tau_1}{\tau_1\tau}\right)} = e^{\frac{-1}{11\tau}t}.$$

(2)

Therefore, the ratio of $E$ to $E_1$ is not only related to $\tau$ but also related to $t$, both nonlinear. In our 3-D model, it will be much more complex, as $\tau$ is changing in different hours, and other processes are involved. Equation (2) provides some qualitative information:

$$\tau > \tau_1, \quad \frac{E}{E_1} > 1$$
$$\tau < \tau_1, \quad \frac{E}{E_1} < 1 \qquad (3)$$

If we reduce $E_0$ by 15%, and the chemical lifetime is $\tau'$.

$$E' = 0.85 E_0 e^{-\frac{1}{\tau'}t}$$

$$\frac{E - E'}{E} = \frac{E_0 e^{-\frac{1}{\tau}t} - 0.85 E_0 e^{-\frac{1}{\tau'}t}}{E_0 e^{-\frac{1}{\tau}t}} = 1 - 0.85 e^{t\left(\frac{\tau'-\tau}{\tau'\tau}\right)}$$

$$if \ \tau' < \tau \qquad (4)$$

$$\frac{E - E'}{E} > 1 - 0.85 = 0.15$$

$$\frac{0.15}{\frac{E - E'}{E}} < 1$$

This is why β and γ values are < 1 at 13:00 − 14:00 when the chemical lifetimes of $NO_x$

in bin #9 in Figure S7 decrease due to decreased anthropogenic $NO_x$ emissions.

Now we assume $\tau' = 0.9\tau$,

$$\frac{E - E'}{E} = 1 - 0.85 e^{t\left(\frac{\tau'-\tau}{\tau'\tau}\right)} = 1 - 0.85 e^{-t\frac{1}{\tau}} \qquad (5)$$

$$0.15 < \frac{E - E'}{E} < 1$$

The left-hand term $\dfrac{E - E'}{E}$ is negatively correlated to $\tau$. With a larger $\tau$, we will have a smaller left-hand term, and then larger $\beta$ and $\gamma$ values. So here, we qualitatively explained your last question: why that $\beta$ and $\gamma$ at 13:00 – 14:00 are smaller than those at 10:00 – 11:00 reflects strong chemical nonlinearity at noon than in the morning? The chemical lifetime of $NO_x$ at noontime is shorter than in the morning. More $NO_x$ is oxidized due to stronger chemistry, and less $NO_2$ is left as surface concentrations or $NO_2$ TVCDs — this is what we called chemical nonlinearity.

We corrected some errors and made some modifications in Lines 238 – 239 and 254 – 265 to make it more consistent with Figure S7. To take into consideration the accumulation of $NO_x$ emissions (several hours of chemical lifetimes) against chemistry, we used the chemical lifetimes at 8:00 – 11:00 and at 11:00 – 14:00, which we think more accurately represent the responses of $NO_2$ TVCD and $NO_2$ surface concentrations to $NO_x$ emissions due to chemical nonlinearity.

• *Abstract, line 17: "non-linearity in the emission-TVCD relationship" should be "anthropogenic emission"*

**Reply:**

Thanks. We corrected it. Please see Line 18 in the revised manuscript.

• *Introduction, line 31: "unfavourable to climate change" – please rephrase*

**Reply:**

Thanks. We changed "which are unfavorable to human health, ecosystem stabilities, and climate change" to ", all of which have negative environmental impacts". Please see Lines 31 - 33 in the revised manuscript.

*• Introduction, line 35: now soil emissions are mentioned specifically making the statement more correct but highlighting that these numbers without an estimate for lightning NOx are incomplete. Please add an estimate for lightning.*

**Reply:**

We added the estimated lightning $NO_x$ emissions over the US in 2014 from the GEOS-Chem model results. Please see Lines 36 – 39 in the revised manuscript.

*• Line 64: paragraph starts with chemical non-linearities, suggesting that the following discussion is about chemistry while I would argue that most of the following observations are explained by the relative contribution of anthropogenic emissions, not chemical non-linearities.*

**Reply:**

Thank you for your suggestion. We changed "the nonlinearity in $NO_x$ chemistry" to "their nonlinear dependences on anthropogenic $NO_x$ emissions". Please see Lines 67 – 68 in the revised manuscript.

*• Line 84: again, chemical non-linearity is mentioned explicitly but I find this misleading.*

**Reply:**

Thanks. We added background sources and physical processes in the sentence. Please see Line 87 in the revised manuscript.

*• Line 204: "in part" – are there any other possible reasons for the non-linearity?*

**Reply:**

Biomass burning is another $NO_x$ source, but its emissions are low and can be neglected over the CONUS compared to lightning and soil $NO_x$ (EPA, 2018; Silvern et al., 2019). Also, biomass burning is mainly in rural regions, and its effects are limited over urban regions in the long term, although severe wildfires may affect urban regions in some specific conditions. Since we used "background sources" in Lines 207 – 208 and biomass burning emissions are also $NO_x$ background sources, we deleted "in part" in Line 207 in the revised manuscript. Also, we added $NO_2$ hydrolysis on aerosols but deleted $NO_2$ wet deposition in Lines 207 – 208. REAM doesn't consider the direct wet deposition of $NO_2$. Therefore, we also updated the sentences in Lines 238 – 239 in the revised manuscript and Lines 68 and 174 in the revised supplement figure file.

*• Line 251: I can't really see the difference in variability between the two overpass times…*

**Reply:**

In Lines 254 – 257 in the revised manuscript, we mean the standard deviations of $\beta$ and $\gamma$ values in the same bins. We changed the sentence to make it clearer. We listed their standard deviations in the following table, clearly showing larger standard deviations at 10:00 − 11:00 LT than 13:00 − 14:00 LT except for β values for bin #1 and bin #8. It is noteworthy that only 1 grid cell belongs to bin #9.

**Table 1.** Uncertainties of β and γ values during different periods for each anthropogenic-$NO_x$-emission bin in Figure 2 in the main manuscript

| | β | | γ | |
| --- | --- | --- | --- | --- |
| | 10:00 - 11:00 | 13:00 - 14:00 | 10:00 - 11:00 | 13:00 - 14:00 |
| bin #1[1] | 17.06 | 18.23 | 15.08 | 14.32 |
| bin #2 | 7.53 | 3.49 | 5.48 | 1.90 |
| bin #3 | 4.83 | 1.84 | 3.64 | 1.09 |
| bin #4 | 3.55 | 1.05 | 1.77 | 0.48 |
| bin #5 | 0.72 | 0.54 | 0.46 | 0.28 |
| bin #6 | 0.62 | 0.31 | 0.37 | 0.19 |
| bin #7 | 0.35 | 0.28 | 0.27 | 0.18 |
| bin #8 | 0.13 | 0.15 | 0.11 | 0.08 |
| bin #9 | 0.00 | 0.00 | 0.00 | 0.00 |

[1] bin #1 denotes $E \in (0, 2^1)$, bin #2 denotes $E \in [2^1, 2^2)$, etc.

As mentioned above, it is hard to quantitively separate their contributions to β and γ values due to the interactions among transport, chemistry, aerosol uptake of $NO_2$, and $NO_2$ dry deposition. However, we can make our estimates indirectly. We have shown that the chemical lifetimes of $NO_x$ change little, the uncertainties of the lifetime relative changes are small, and chemical nonlinearity is not a big issue in low-anthropogenic-$NO_x$ emission regions (Figure S7). $NO_2$ hydrolysis on aerosols and dry deposition are proportional to $NO_2$ concentrations which are determined by transport and chemistry. The lifetimes of $NO_x$ against $NO_2$ hydrolysis and dry deposition are almost the same for

"group 1" and "group 2" simulation results. That is to say, transport is the most critical factor in non-emission factors (excluding background sources) in low-anthropogenic-$NO_x$ emission regions. As the uncertainties of $\beta_{Emis}$ at 10:00 – 11:00 LT are close to those at 13:00 – 14:00 LT (their relative differences are < 15%), and the uncertainties of $\gamma_{Emis}$ are the same for 10:00 – 11:00 and 13:00 – 14:00 LT, the differences of the standard deviations of $\beta$ ($\gamma$) values at 10:00 – 11:00 from those at 13:00 – 14:00 are mainly from non-emission factors — that is transport dominated in low-anthropogenic-$NO_x$ emission regions.

• *Line 256: … and even if it were present, why is that indicative of chemical non-linearity?*

**Reply:**

Please see the answer to the first question. Also, in Figure S7, (g) and (h) shows that the relative changes of $NO_x$ chemical lifetime at noontime are even larger than those in the morning, again causing $\beta$ and $\gamma$ values at noontime smaller than in the morning.

**References**

[revised manuscript text omitted]

**Table Captions**

Table S1. Summary of major satellite instruments for remote sensing of atmospheric $NO_2$ VCD in the past decade

Table S2. Summary of satellite $NO_2$ TVCD products and their retrieval information

Table S3. Selection criteria for satellite $NO_2$ TVCD pixel data

Table S4. Summary of annual trends of AQS $NO_2$ surface concentrations and satellite $NO_2$ TVCD products in each region during different periods

**Table S1. Summary of major satellite instruments for remote sensing of atmospheric NO₂ VCD in the past decade**

| Instrument | Satellite | Launch date | End date | Operator | Equator crossing time (local time) | UV/Vis Spectral range (nm) | Spectral resolution (nm) | Swath length (km) | Nadir pixel resolution (km × km) | Global coverage (days) |
|---|---|---|---|---|---|---|---|---|---|---|
| SCIAMACHY | ENVISAT[1] | 03/01/2002[2] | 04/08/2012[2] | ESA[3] | 10:00[1] | $240 - 805$[4] | $0.24 - 0.48$[4] | 960[5] | $60 \times 30$[5] | 6[5] |
| GOME-2A | MetOp-A[6] | 10/19/2006[6] | in operation | EUMETSAT[7] | 9:30[8] | $240 - 790$[8] | $0.26 - 0.51$[8] | 1920 before Jul. 15th, 2013; 960 after Jul. 15th, 2013[8] | $80 \times 40$ before Jul. 15th, 2013; $40 \times 40$ after Jul. 15th, 2013[8] | 1.5[9] |
| GOME-2B | MetOp-B[6] | 09/17/2012[6] | In operation | EUMETSAT | 9:30[8] | $240 - 790$[8] | $0.26 - 0.51$[8] | 1920[8] | $80 \times 40$[8] | 1.5[9] |
| OMI | EOS-Aura[10] | 07/152004[10] | In operation | NASA | 13:45[10] | $270 - 500$[11] | $0.45 - 1.0$[11] | 2600[11] | $24 \times 13$[11] | 1[11] |

[1] Refer to https://earth.esa.int/web/guest/missions/esa-operational-eo-missions/envisat
[2] Refer to https://en.wikipedia.org/wiki/Envisat
[3] The European Space Agency
[4] Refer to http://www.iup.uni-bremen.de/sciamachy/instrument/performance/index.html
[5] Refer to Boersma et al. (2008), Boersma et al. (2009), and Lee et al. (2009)
[6] Refer to https://www.eumetsat.int/website/home/Satellites/CurrentSatellites/Metop/index.html
[7] The European Organization for the Exploitation of Meteorological Satellites
[8] Refer to EUMETSAT (2015)
[9] Refer to Lee et al. (2009) and Wang et al. (2017)
[10] Refer to https://aura.gsfc.nasa.gov/
[11] Refer to https://aura.gsfc.nasa.gov/omi.html

**Table S2. Summary of satellite NO₂ TVCD products and their retrieval information**

| NO$_2$ TVCD products | Version | Available period | DOAS fitting method | Stratosphere–troposphere separation | Fitting window (nm) | Albedo / reflectance | A priori profiles | Radiative transfer model | Cloud | Uncertainty |
|---|---|---|---|---|---|---|---|---|---|---|
| GOME-2B | TM4NO2A (2.3) | 12/20/2012 – current | Intensity fit[1] | Assimilation of satellite total slant columns in the TM4 model[2,3] | 405 – 465[1] | Climatology albedo from 3 years of OMI data[4] | TM4 (2° × 3°)[2] | DAK[2] | FRESCO+ (Oxygen A-band around 760 nm)[5] | 1.0 × 10$^{15}$ molecules/cm$^2$ + 25%[2] |
| SCIAMACHY | QA4ECV (v1.1) | 08/02/2002 – 04/08/2012 | Optical Density[1,6] | Assimilation of OMI total slant columns in the TM5 - MP model[6,7] | 425 – 465[6] | Climatology albedo based on SCIAMACHY[8] | TM5-MP (1° × 1°)[6] | DAK | FRESCO+ | 35% - 45% over polluted scenes; > 100% over background regions (Pacific Ocean)[6] |
| GOME-2A | QA4ECV (v1.1) | 02/01/2007 – 12/31/2016 | | | 405 – 465[1,6] | Climatology albedo based on GOME-2A[8] | | | FRESCO+ | |
| OMI-QA4ECV | QA4ECV (v1.1) | 10/012004 – Current | | | 405 – 465[1,6] | Climatology albedo from 5 years of OMI data[6] | | | Improved O$_2$-O$_2$ (477 nm)[9] | |
| OMI-NASA | SPv3 | 01/01/2005 – 07/31/2017 | Stepwise intensity fit with monthly averaged solar irradiance spectrum[1,10] | Based on OMI total slant columns over regions with low estimated TVCD contributions (TVCD contributions less than 0.3 × 10$^{15}$ molecules/cm$^2$)[10] | 402 – 465[1,10] | OMI climatology albedo[10] | GMI (1° × 1.25°)[10] | TMORAD[10] | O$_2$-O$_2$ (477 nm)[10,11] | SPv2.1 TVCD has uncertainties of about 30% under clear-sky conditions to about 60% under cloudy conditions[12], and the relative difference between SPv3 and SPv2.1 is less than ~20%[10]. |
| OMI-BEHR[13] | v3.0B | 01/01/2005 – 07/31/2017 | | | | Based on MCD43D BRDF product (for land) and model parameterization (for ocean) | WRF-Chem (12 km) | | | ~ 45% on average[14] |

[1] Refer to Zara et al. (2018)
[2] Refer to Boersma et al. (2011). "TM4" is the Tracer Model, version 4. "DAK" is the Doubling-Adding KNMI (DAK) radiative transfer model.
[3] Refer to Williams et al. (2009)
[4] Refer to Kleipool et al. (2008)
[5] Refer to Wang et al. (2017) and Wang et al. (2008)
[6] Refer to Boersma et al. (2018)
[7] Refer to Williams et al. (2017)
[8] Refer to Tilstra et al. (2017)
[9] Refer to Veefkind et al. (2016)
[10] Refer to Bucsela et al. (2013), Bucsela et al. (2016), Krotkov et al. (2017), and Marchenko et al. (2015). "TMORAD" is the TMOS radiative transfer model.
[11] Refer to Acarreta et al. (2004)
[12] Refer to Lamsal et al. (2014), Oetjen et al. (2013), and Tong et al. (2015)
[13] Refer to Laughner et al. (2018). OMI-BEHR uses the SCD from OMI-NASA SPv3 but updates inputs for the AMF calculation, such as a prior NO₂ vertical profiles and surface reflectance. Besides, OMI-BEHR only provides NO₂ TVCD over the contiguous
United States (CONUS). As in this study, we used the OMI-NASA datasets archived in the OMI-BEHR product, so we only obtained OMI-NASA datasets extended to July 31, 2017.
[14] Average uncertainty over the CONUS is calculated based on the file from http://behr.cchem.berkeley.edu/behr/BEHR-us-uncertainty.hdf

**Table S3. Selection criteria for satellite NO$_2$ TVCD pixel data**

| NO$_2$ TVCD products | Period | Solar zenith angle | albedo | Cloud radiance fraction | Snow or ice covered | AMFtrop/AMFgeo | Flag for retrieval success | Retrieval quality flag | Rows in swath |
|---|---|---|---|---|---|---|---|---|---|
| GOME-2B | 01/01/2013 – 12/31/2017 | < 80° | <= 0.3 | <= 50% | No | > 0.2 | Yes | | All |
| SCIAMACHY | 01/01/2003 – 12/31/2011 | < 80° | <= 0.3 | <= 50% | No | > 0.2 | Yes | | All |
| GOME-2A | 01/01/2008 – 12/31/2016 | < 80° | <= 0.3 | <= 50% | No | > 0.2 | Yes | | All |
| OMI-QA4ECV[1] | 01/01/2005 – 12/31/2017 | < 80° | <= 0.3 | <= 50% | No | > 0.2 | Yes | | 6 - 21 |
| OMI–NASA[1] | 01/01/2005 – 12/31/2016 | < 80° | <= 0.3 | <= 50% | | | Yes | Yes | 6 – 21 |
| OMI-BEHR[1] | 01/01/2005 – 12/31/2016 | < 80° | <= 0.3 | <= 50% | | | Yes | Yes | 6 - 21 |

[1] Rows 6-21 are selected to remove the anomalies developed in the OMI sensor (Boersma et al., 2018; Zhang et al., 2018).

**Table S4. Summary of annual trends of AQS NO$_2$ surface concentrations and satellite NO$_2$ TVCD products in each region during different periods[1]**

| | | Northeast | | Midwest | | South | | West | |
|---|---|---|---|---|---|---|---|---|---|
| | | AQS site | CONUS | AQS site | CONUS | AQS site | CONUS | AQS site | CONUS |
| AQS NO$_2$ VMR at 13:00 -14:00 | 2003 – 2011 | -6.8 ± 0.7% | | -6.1 ± 1.2% | | -6.6 ± 0.7% | | -7.6 ± 1.2% | |
| | 2011 – 2017 | -8.0 ± 1.2% | | -6.4 ± 0.8% | | -5.8 ± 0.6% | | -7.2 ± 1.6% | |
| AQS NO$_2$ VMR at 10:00 – 11:00 | 2003 – 2011 | -6.6 ± 0.5% | | -5.8 ± 1.5% | | -6.5 ± 1.3% | | -7.1 ± 1.6% | |
| | 2011 – 2017 | -7.6 ± 1.0% | | -6.8 ± 0.5% | | -5.7 ± 0.1% | | -6.1 ± 1.1% | |
| SCIAMACHY | 2003 – 2011 | -17.1 ± 2.7% | -11.0 ± 3.3% | -12.9 ± 6.8% | -6.5 ± 0.8% | -9.1 ± 1.0% | -6.2 ± 1.5% | -9.1 ± 1.8% | -7.0 ± 1.4% |
| | 2011 – 2017 | | | | | | | | |
| GOME2B | 2003 – 2011 | | | | | | | | |
| | 2013 – 2017 | -11.4 ± 3.7% | -10.8 ± 3.9% | -9.9 ± 13.1% | -4.4 ± 27.2% | -8.9 ± 3.0% | -7.5 ± 3.6% | -11.8 ± 3.0% | -10.6 ± 2.3% |
| OMI-QA4ECV | 2005 – 2011 | -14.2 ± 6.3% | -10.6 ± 3.8% | -9.2 ± 4.2% | -8.4 ± 2.8% | -9.2 ± 2.7% | -8.2 ± 1.5% | -10.5 ± 1.6% | -8.7 ± 0.9% |
| | 2011 – 2017 | -18.0 ± 16.2% | -7.6 ± 4.2% | -7.6 ± 3.3% | -7.0 ± 1.7% | -4.8 ± 1.4% | -4.6 ± 1.0% | -6.4 ± 1.4% | -4.8 ± 1.2% |
| OMI-NASA | 2005 – 2011 | -11.8 ± 1.3% | -11.0 ± 1.8% | -10.9 ± 4.8% | -10.0 ± 4.1% | -10.0 ± 3.5% | -9.5 ± 1.9% | -10.2 ± 1.8% | -8.5 ± 0.9% |
| | 2011 – 2016 | -10.0 ± 4.9% | -8.5 ± 3.8% | -13.2 ± 3.2% | -9.2 ± 2.7% | 0.3 ± 19.2% | -8.0 ± 5.5% | -9.0 ± 5.7% | -6.6 ± 3.9% |
| OMI-BEHR | 2005 – 2011 | -11.8 ± 1.8% | -10.9 ± 1.9% | -12.2 ± 7.3% | -9.8 ± 4.4% | -9.5 ± 3.1% | -8.8 ± 2.0% | -9.9 ± 1.1% | -8.2 ± 0.4% |
| | 2011 – 2016 | -8.2 ± 3.4% | -6.6 ± 1.7% | -27.4 ± 24.3% | -8.1 ± 3.0% | -7.2 ± 2.3% | -5.0 ± 1.3% | -13.2 ± 14.5% | -7.0 ± 4.8% |

[1] Annual trends are the averages of regional seasonal trends (e.g, Figure 7).

[*] *Correspondence to* Yuhang Wang (yuhang.wang@eas.gatech.edu)

**Figure Captions**

Figure S1. Annual variation of $NO_3^-$ wet deposition fluxes for each season from 2003 – 2017. The fluxes were scaled by the corresponding values in 2003. Shaded regions denote standard deviations. Monthly $NO_3^-$ wet deposition observations are obtained from https://nadp.slh.wisc.edu/data/NTN/ntnAllsites.aspx (last access, September 29, 2019).

Figure S2. Comparison between original EPA anthropogenic $NO_x$ emissions and updated EPA anthropogenic $NO_x$ emissions with the newest Continuous Emission Monitoring Systems (CEMS) measurements.

Figure S3. Daily OMI $NO_2$ TVCDs for July 2011 (a) and 2012 (b) in Atlanta (33.755° N, 84.39° W). Black circles are weekday values, and red circles are weekend values. We find significant daily variations of $NO_2$ TVCD from (a) and (b). The number of available measurements in July 2011 is much less than July 2012. We find clear larger $NO_2$ TVCD values on weekdays than on weekends in July 2011, but the difference between weekday and weekday TVCDs in July 2012 are not so obvious.

Figure S4. Hourly averaged ratios of FEM (a) and CAPS (b) to FRM $NO_2$ measurements in each season, respectively. The FEM/FRM ratios are computed from coincident FRM and FEM measurements from 2013 – 2015 at 4 sites. The CAPS/FRM ratios are calculated based on coincident CAPS and FRM data from 2015 – 2016 at 3 sites.

Figure S5. Annual variations of AQS $NO_2$ surface concentrations at different hours on weekdays in spring (a, b), summer (c, d), autumn (e, f), and winter (g, h). Left panels show absolute $NO_2$ concentrations, and right panels are their relative variations normalized to 2011. To conduct reliable and consistent comparisons, we only used monitoring sites satisfying the seasonal *RCI* < 50% and continuity criteria on weekdays from 2003 – 2017.

Figure S6. Distributions of (a) $NO_2$ TVCD fraction that is in the boundary layer (< 2810 m) at

13:00 – 14:00, (b) $NO_2$ TVCD fraction in the boundary layer (< 1290 m) at 10:00 – 11:00, (c) the fraction of soil $NO_x$ emissions in all surface sources (anthropogenic + soil) on weekdays for July

2011. As the lifetime of $NO_2$ in the free troposphere (several days ~ 2 weeks) is much longer than that in the boundary layer (~ 10 hours), local lightning $NO_x$ emissions cannot represent $NO_2$

VCDs in the free troposphere. In this study, we apply $NO_2$ VCD in the free troposphere to analyze the impact of lighting $NO_x$ on the nonlinear relationships between anthropogenic $NO_x$

emissions and $NO_2$ TVCDs and use lightning $NO_x$ and $NO_2$ VCD in the free troposphere interchangeably in the following.

Figure S7. (a) Distributions of the fractions of surface $NO_x$ emissions emitted by soil ("SoilNO$_x$"), the portions of $NO_2$ TVCDs in the boundary layer ("PBLVCD"), and the fractions of $NO_2$ TVCDs from anthropogenic $NO_x$ emissions ("AnthroVCD") as functions of NEI2011

anthropogenic $NO_x$ emissions at 13:00 – 14:00 LT on weekdays for July 2011 over the CONUS.

The fraction of $NO_2$ TVCDs from anthropogenic $NO_x$ emissions is equal to $\left(1 - \right.$

$\left. \frac{E_{soil}}{E_{soil}+E_{anthropogenic}}\right) \times \left(\frac{TVCD_{boundary}}{TVCD_{boundary}+TVCD_{free}}\right)$, where $E_{soil}$ denotes soil $NO_x$ emissions,

$E_{anthropogenic}$ denotes anthropogenic $NO_x$ emissions, $TVCD_{boundary}$ denotes $NO_2$ TVCDs in the boundary layer, and $TVCD_{free}$ denotes $NO_2$ TVCDs in the free troposphere. The calculated data are grouped into 9 bins as in Figure 2. (b) Same as (a), but for 10:00 – 11:00 LT. (c) Distributions of $\beta_{Emis}$, $\gamma_{Emis}$, $\beta$, and $\gamma$ as functions of anthropogenic $NO_x$ emissions at 13:00 – 14:00 LT on weekdays for July 2011 over the CONUS. $\beta$ and $\gamma$ are the same as Figure 2. $\beta_{Emis}$ and $\gamma_{Emis}$ denote

$\beta$ and $\gamma$ values when no other factors are taken into consideration except for soil $NO_x$ emissions, anthropogenic $NO_x$ emissions, and $NO_2$ in the free troposphere. $\beta_{Emis} =$

$\frac{15\%}{15\% \times \left(\frac{E_{anthropogenic}}{E_{anthropogenic}+E_{soil}}\right)\left(\frac{TVCD_{boundary}}{TVCD_{boudnary}+TVCD_{free}}\right)} = \left(\frac{E_{anthropogenic}+E_{soil}}{E_{anthropogenic}}\right)\left(\frac{TVCD_{boundary}+TVCD_{free}}{TVCD_{boundary}}\right),$

and $\gamma_{Emis} = \dfrac{15\%}{15\% \times \left(\frac{E_{anthropogenic}}{E_{anthropogenic}+E_{soil}}\right)} = \left(\dfrac{E_{anthropogenic}+E_{soil}}{E_{anthropogenic}}\right)$. It is noteworthy that here we assume no interactions between the boundary layer and the free troposphere, boundary-layer $NO_x$

are only related to soil and anthropogenic $NO_x$ emissions, and lightning $NO_x$ only affect $NO_2$ in the free troposphere. The assumptions are reasonable as the time scale ($\sim$ 1 week) of the interactions between the boundary layer and the free troposphere  is much longer than $NO_x$

lifetime in the boundary layer, and  only a small fraction of lightning $NO_x$ is distributed into the boundary layer in this study. Therefore, $\beta_{Emis}$ and $\gamma_{Emis}$ roughly represent the contributions of background sources (lightning $NO_x$ and soil $NO_x$) to $\beta$ and $\gamma$ values. The differences between $\beta$ ($\gamma$) and $\beta_{Emis}$ ($\gamma_{Emis}$) indicate the contribution of non-emission factors to $\beta$

($\gamma$) values, such as chemistry, transport, $NO_2$ hydrolysis on aerosols, and dry  deposition.

(d) Same as (c), but for 10:00 – 11:00 LT. From (c) and (d), we find that both background sources (lightning $NO_x$ + soil $NO_x$) and non-emission factors are important when considering the nonlinear relationships among $NO_x$ emissions, $NO_2$ surface concentrations, and

$NO_2$ TVCDs in low-anthropogenic-$NO_x$ emission regions. (e) Distribution of $NO_x$ chemical lifetimes as functions of anthropogenic $NO_x$ emissions at 11:00 – 14:00 LT on weekdays for July

2011 over the CONUS. "Standard_surf" denotes $NO_x$ chemical lifetimes at the surface layer from the standard REAM simulation ("group 1" in Section 3.1); "Standard_trop" denotes average $NO_x$

chemical lifetimes in the troposphere for "group 1"; "Reduce_surf" denotes $NO_x$ chemical lifetimes at the surface layer for "group 2" with anthropogenic $NO_x$ emissions reduced by 15%;

"Reduce_trop" denotes average $NO_x$ chemical lifetimes in the troposphere for "group 2". In this study, we used the lifetimes at 11:00 – 14:00 LT but not 13:00 – 14:00 LT to partly include the accumulation effect of $NO_x$ emissions: $NO_2$ TVCD and $NO_2$ surface concentrations at 13:00 –

14:00 LT are not only affected by $NO_x$ emissions at 13:00 – 14:00 LT but also by $NO_x$ emissions before that due to the $NO_x$ chemical lifetime of several hours in daytime. (f) Same as (e), but for

8:00 – 11:00 LT. (g) Relative changes of $NO_x$ chemical lifetimes at 11:00 – 14:00 LT on weekdays for July 2011 over the CONUS due to the 15% decrease of anthropogenic $NO_x$

emissions in "group 2". "Surface" denotes the relative changes of $NO_x$ chemical lifetimes at the surface, while "Troposphere" denotes the relative changes of average $NO_x$ chemical lifetimes in the troposphere. We first calculated the relative changes in each grid cell via $\frac{lifetime_{Reduce}}{lifetime_{Standard}} - 1$, and then binned the calculated data into 9 groups as Figure 2. (h) Same as (g), but for 8:00 –

11:00 LT. In the chemical lifetime calculation, we included sinks from the reaction of $OH + NO_2$

and net losses due to organic nitrate production from the reactions of $RO_2$ with NO or $NO_2$ except for peroxyacyl nitrates (PANs), because PANs can be either a source or sink of $NO_x$ depending on transport and chemistry. Only accounting for the sink from the reaction of $OH + NO_2$ produces significant different lifetimes in low-anthropogenic-$NO_x$ emission bins and has less impact on high-anthropogenic-$NO_x$ emission regions, which, however, does not affect our conclusions derived from subpanels (g) and (h) (the mean relative differences of chemical lifetimes between

"group 1" and "group 2" are still < 10% in all bins): the chemical nonlinearity contributes little to

β and γ values in low-anthropogenic-$NO_x$ emission regions. Although not shown here, the impacts of $NO_2$ hydrolysis and $NO_2$ dry deposition on β and γ values are even smaller than those of chemical nonlinearity. Therefore, the differences between β (γ) and $β_{Emis}$ ($γ_{Emis}$) in low- anthropogenic-$NO_x$ emission bins in (c) and (d) mainly indicate the contribution of transport to β

(γ) values. Error bars in (a), (b), (g), and (h) denote standard deviations.

Figure S8. Same as Figure 4, but for AQS $NO_2$ surface concentrations and coincident GOME-2A

$NO_2$ TVCD data during 2008 – 2016.

Figure S9. Relative variations of OMI-QA4ECV $NO_2$ TVCD data for urban regions (black lines)

and the whole CONUS (red lines) from 2005 – 2017 in 4 seasons.

[Figure]

Figure S1. Annual variation of $NO_3^-$ wet deposition fluxes for each season from 2003 – 2017. The fluxes were scaled by the corresponding values in 2003. Shaded regions denote standard deviations. Monthly $NO_3^-$ wet deposition observations are obtained from https://nadp.slh.wisc.edu/data/NTN/ntnAllsites.aspx (last access, September 29, 2019).

[Figure]

Figure S2. Comparison between original EPA anthropogenic $NO_x$ emissions and updated EPA

anthropogenic $NO_x$ emissions with the newest Continuous Emission Monitoring Systems (CEMS) measurements.

[Figure]

Figure S3. Daily OMI $NO_2$ TVCDs for July 2011 (a) and 2012 (b) in Atlanta (33.755° N, 84.39°

W). Black circles are weekday values, and red circles are weekend values. We find significant daily variations of $NO_2$ TVCD from (a) and (b). The number of available measurements in July

2011 is much less than July 2012. We find clear larger $NO_2$ TVCD values on weekdays than on weekends in July 2011, but the difference between weekday and weekday TVCDs in July 2012

are not so obvious.

[Figure]

Figure S4. Hourly averaged ratios of FEM (a) and CAPS (b) to FRM NO$_2$ measurements in each season, respectively. The FEM/FRM ratios are computed from coincident FRM and FEM

measurements from 2013 – 2015 at 4 sites. The CAPS/FRM ratios are calculated based on coincident CAPS and FRM data from 2015 – 2016 at 3 sites.

[Figure]

Figure S5. Annual variations of AQS $NO_2$ surface concentrations at different hours on weekdays in spring (a, b), summer (c, d), autumn (e, f), and winter (g, h). Left panels show absolute $NO_2$

concentrations, and right panels are their relative variations normalized to 2011. To conduct reliable and consistent comparisons, we only used monitoring sites satisfying the seasonal *RCI* <

50% and continuity criteria on weekdays from 2003 – 2017.

[Figure]

Figure S6. Distributions of (a) $NO_2$ TVCD fraction that is in the boundary layer (< 2810 m) at

13:00 − 14:00, (b) $NO_2$ TVCD fraction in the boundary layer (< 1290 m) at 10:00 − 11:00, (c) the fraction of soil $NO_x$ emissions in all surface sources (anthropogenic + soil) on weekdays for July

2011. As the lifetime of $NO_2$ in the free troposphere (several days ~ 2 weeks) is much longer than that in the boundary layer (~ 10 hours), local lightning $NO_x$ emissions cannot represent $NO_2$

VCDs in the free troposphere. In this study, we apply $NO_2$ VCD in the free troposphere to analyze the impact of lighting $NO_x$ on the nonlinear relationships between anthropogenic $NO_x$

emissions and $NO_2$ TVCDs and use lightning $NO_x$ and $NO_2$ VCD in the free troposphere interchangeably in the following.

[Figure]

[Figure]

Figure S7. (a) Distributions of the fractions of surface NO$_x$ emissions emitted by soil ("SoilNO$_x$"), the portions of NO$_2$ TVCDs in the boundary layer ("PBLVCD"), and the fractions of NO$_2$ TVCDs from anthropogenic NO$_x$ emissions ("AnthroVCD") as functions of NEI2011

anthropogenic NO$_x$ emissions at 13:00 − 14:00 LT on weekdays for July 2011 over the CONUS.

The fraction of NO$_2$ TVCDs from anthropogenic NO$_x$ emissions is equal to $\left(1 - \right.$

$\left.\dfrac{E_{soil}}{E_{soil}+E_{anthropogenic}}\right) \times \left(\dfrac{TVCD_{boundary}}{TVCD_{boundary}+TVCD_{free}}\right)$, where $E_{soil}$ denotes soil NO$_x$ emissions,

$E_{anthropogenic}$ denotes anthropogenic $NO_x$ emissions, $TVCD_{boundary}$ denotes $NO_2$ TVCDs in the boundary layer, and $TVCD_{free}$ denotes $NO_2$ TVCDs in the free troposphere. The calculated data are grouped into 9 bins as in Figure 2. (b) Same as (a), but for 10:00 – 11:00 LT. (c) Distributions of $\beta_{Emis}$, $\gamma_{Emis}$, $\beta$, and $\gamma$ as functions of anthropogenic $NO_x$ emissions at 13:00 – 14:00 LT on weekdays for July 2011 over the CONUS. $\beta$ and $\gamma$ are the same as Figure 2. $\beta_{Emis}$ and $\gamma_{Emis}$ denote

$\beta$ and $\gamma$ values when no other factors are taken into consideration except for soil $NO_x$ emissions, anthropogenic $NO_x$ emissions, and $NO_2$ in the free troposphere. $\beta_{Emis} =$

$$\frac{15\%}{15\% \times \left(\frac{E_{anthropogenic}}{E_{anthropogenic}+E_{soil}}\right)\left(\frac{TVCD_{boundary}}{TVCD_{boudnary}+TVCD_{free}}\right)} = \left(\frac{E_{anthropogenic}+E_{soil}}{E_{anthropogenic}}\right)\left(\frac{TVCD_{boundary}+TVCD_{free}}{TVCD_{boundary}}\right),$$

and $\gamma_{Emis} = \dfrac{15\%}{15\% \times \left(\frac{E_{anthropogenic}}{E_{anthropogenic}+E_{soil}}\right)} = \left(\dfrac{E_{anthropogenic}+E_{soil}}{E_{anthropogenic}}\right)$. It is noteworthy that here we assume no interactions between the boundary layer and the free troposphere, boundary-layer $NO_x$

are only related to soil and anthropogenic $NO_x$ emissions, and lightning $NO_x$ only affect $NO_2$ in the free troposphere. The assumptions are reasonable as the time scale (~ 1 week) of the interactions between the boundary layer and the free troposphere  is much longer than $NO_x$

lifetime in the boundary layer, and  only a small fraction of lightning $NO_x$ is distributed into the boundary layer in this study. Therefore, $\beta_{Emis}$ and $\gamma_{Emis}$ roughly represent the contributions of background sources (lightning $NO_x$ and soil $NO_x$) to $\beta$ and $\gamma$ values. The differences between $\beta$ ($\gamma$) and $\beta_{Emis}$ ($\gamma_{Emis}$ ) indicate the contribution of non-emission factors to $\beta$

($\gamma$) values, such as chemistry, transport, $NO_2$ hydrolysis on aerosols, and dry  deposition.

(d) Same as (c), but for 10:00 – 11:00 LT. From (c) and (d), we find that both background sources (lightning $NO_x$ + soil $NO_x$) and non-emission factors are important when considering the nonlinear relationships among $NO_x$ emissions, $NO_2$ surface concentrations, and

$NO_2$ TVCDs in low-anthropogenic-$NO_x$ emission regions. (e) Distribution of $NO_x$ chemical lifetimes as functions of anthropogenic $NO_x$ emissions at 11:00 – 14:00 LT on weekdays for July

2011 over the CONUS. "Standard_surf" denotes $NO_x$ chemical lifetimes at the surface layer from the standard REAM simulation ("group 1" in Section 3.1); "Standard_trop" denotes average $NO_x$

chemical lifetimes in the troposphere for "group 1"; "Reduce_surf" denotes $NO_x$ chemical lifetimes at the surface layer for "group 2" with anthropogenic $NO_x$ emissions reduced by 15%;

"Reduce_trop" denotes average $NO_x$ chemical lifetimes in the troposphere for "group 2". In this study, we used the lifetimes at 11:00 – 14:00 LT but not 13:00 – 14:00 LT to partly include the accumulation effect of $NO_x$ emissions: $NO_2$ TVCD and $NO_2$ surface concentrations at 13:00 –

14:00 LT are not only affected by $NO_x$ emissions at 13:00 – 14:00 LT but also by $NO_x$ emissions before that due to the $NO_x$ chemical lifetime of several hours in daytime. (f) Same as (e), but for

8:00 – 11:00 LT. (g) Relative changes of $NO_x$ chemical lifetimes at 11:00 – 14:00 LT on weekdays for July 2011 over the CONUS due to the 15% decrease of anthropogenic $NO_x$

emissions in "group 2". "Surface" denotes the relative changes of $NO_x$ chemical lifetimes at the surface, while "Troposphere" denotes the relative changes of average $NO_x$ chemical lifetimes in the troposphere. We first calculated the relative changes in each grid cell via $\frac{lifetime_{Reduce}}{lifetime_{Standard}} - 1$, and then binned the calculated data into 9 groups as Figure 2. (h) Same as (g), but for 8:00 –

11:00 LT. In the chemical lifetime calculation, we included sinks from the reaction of OH + $NO_2$

and net losses due to organic nitrate production from the reactions of $RO_2$ with NO or $NO_2$ except for peroxyacyl nitrates (PANs), because PANs can be either a source or sink of $NO_x$ depending on transport and chemistry. Only accounting for the sink from the reaction of OH + $NO_2$ produces significant different lifetimes in low-anthropogenic-$NO_x$ emission bins and has less impact on high-anthropogenic-$NO_x$ emission regions, which, however, does not affect our conclusions derived from subpanels (g) and (h) (the mean relative differences of chemical lifetimes between

"group 1" and "group 2" are still < 10% in all bins): the chemical nonlinearity contributes little to

$\beta$ and $\gamma$ values in low-anthropogenic-$NO_x$ emission regions. Although not shown here, the impacts of $NO_2$ hydrolysis and $NO_2$ dry deposition on $\beta$ and $\gamma$ values are even smaller than those of chemical nonlinearity. Therefore, the differences between $\beta$ ($\gamma$) and $\beta_{Emis}$ ($\gamma_{Emis}$) in low- anthropogenic-$NO_x$ emission bins in (c) and (d) mainly indicate the contribution of transport to $\beta$

($\gamma$) values. Error bars in (a), (b), (g), and (h) denote standard deviations.

[Figure]

Figure S8. Same as Figure 4, but for AQS NO$_2$ surface concentrations and coincident GOME-2A

NO$_2$ TVCD data during 2008 – 2016.

[Figure]

Figure S9. Relative variations of OMI-QA4ECV NO$_2$ TVCD data for urban regions (black lines)
and the whole CONUS (red lines) from 2005 – 2017 in 4 seasons.